# Structural characterization of encapsulated ferritin provides insight into iron storage in bacterial nanocompartments

Didi He[1], Sam Hughes[2], Sally Vanden-Hehir[2], Atanas Georgiev[1], Kirsten Altenbach[1], Emma Tarrant[3], C Logan Mackay[2], Kevin J Waldron[3], David J Clarke[2]*, Jon Marles-Wright[1,4]*

[1]Institute of Quantitative Biology, Biochemistry and Biotechnology, School of Biological Sciences, The University of Edinburgh, Edinburgh, United Kingdom; [2]The School of Chemistry, The University of Edinburgh, Edinburgh, United Kingdom; [3]Institute for Cell and Molecular Biosciences, Newcastle University, Newcasle upon Tyne, United Kingdom; [4]School of Biology, Newcastle University, Newcastle upon Tyne, United Kingdom

*For correspondence: dave.
clarke@ed.ac.uk (DJC); jon.marles-
wright1@ncl.ac.uk (JM-W)

Competing interests: The authors declare that no competing interests exist.

**Abstract** Ferritins are ubiquitous proteins that oxidise and store iron within a protein shell to protect cells from oxidative damage. We have characterized the structure and function of a new member of the ferritin superfamily that is sequestered within an encapsulin capsid. We show that this encapsulated ferritin (EncFtn) has two main alpha helices, which assemble in a metal dependent manner to form a ferroxidase center at a dimer interface. EncFtn adopts an open decameric structure that is topologically distinct from other ferritins. While EncFtn acts as a ferroxidase, it cannot mineralize iron. Conversely, the encapsulin shell associates with iron, but is not enzymatically active, and we demonstrate that EncFtn must be housed within the encapsulin for iron storage. This encapsulin nanocompartment is widely distributed in bacteria and archaea and represents a distinct class of iron storage system, where the oxidation and mineralization of iron are distributed between two proteins.

## Introduction

Encapsulin nanocompartments are a family of proteinaceous metabolic compartments that are widely distributed in bacteria and archaea (*Sutter et al., 2008*; *Akita et al., 2007*; *McHugh et al., 2014*; *Contreras et al., 2014*). They share a common architecture, comprising an icosahedral shell formed by the oligomeric assembly of a protein, encapsulin, that is structurally related to the HK97 bacteriophage capsid protein gp5 (*Helgstrand et al., 2003*). Gp5 is known to assemble as a 66 nm diameter icosahedral shell of 420 subunits. In contrast, both the *Pyrococcus furiosus* (*Akita et al., 2007*) and *Myxococcus xanthus* (*McHugh et al., 2014*) encapsulin shell-proteins form 32 nm icosahedra with 180 subunits; while the *Thermotoga maritima* (*Sutter et al., 2008*) encapsulin is smaller still with a 25 nm, 60-subunit icosahedron. The high structural similarity of the encapsulin shell-proteins to gp5 suggests a common evolutionary origin for these proteins (*McHugh et al., 2014*).

The genes encoding encapsulin proteins are found downstream of genes for dye-dependent peroxidase (DyP) family enzymes (*Roberts et al., 2011*), or encapsulin-associated ferritins (EncFtn) (*He and Marles-Wright, 2015*). Enzymes in the DyP family are active against polyphenolic

**eLife digest** Iron is essential for life as it is a key component of many different enzymes that participate in processes such as energy production and metabolism. However, iron can also be highly toxic to cells because it readily reacts with oxygen. This reaction can damage DNA, proteins and the membranes that surround cells.

To balance the cell's need for iron against its potential damaging effects, organisms have evolved iron storage proteins known as ferritins that form cage-like structures. The ferritins convert iron into a less reactive form that is mineralised and safely stored in the central cavity of the ferritin cage and is available for cells when they need it.

Recently, a new family of ferritins known as encapsulated ferritins have been found in some microorganisms. These ferritins are found in bacterial genomes with a gene that codes for a protein cage called an encapsulin. Although the structure of the encapsulin cage is known to look like the shell of a virus, the structure that the encapsulated ferritin itself forms is not known. It is also not clear how encapsulin and the encapsulated ferritin work together to store iron.

He et al. have now used the techniques of X-ray crystallography and mass spectrometry to determine the structure of the encapsulated ferritin found in some bacteria. The encapsulated ferritin forms a ring-shaped doughnut in which ten subunits of ferritin are arranged in a ring; this is totally different from the enclosed cages that other ferritins form.

Biochemical studies revealed that the encapsulated ferritin is able to convert iron into a less reactive form, but it cannot store iron on its own since it does not form a cage. Thus, the encapsulated ferritin needs to be housed within the encapsulin cage to store iron.

Further work is needed to investigate how iron moves into the encapsulin cage to reach the ferritin proteins. Some organisms have both standard ferritin cages and encapsulated ferritins; why this is the case also remains to be discovered.

compounds such as azo dyes and lignin breakdown products; although their physiological function and natural substrates are not known (*Roberts et al., 2011*). Ferritin family proteins are found in all kingdoms and have a wide range of activities, including ribonucleotide reductase (*Aberg et al., 1993*), protecting DNA from oxidative damage (*Grant et al., 1998*), and iron storage (*Bradley et al., 2014*). The classical iron storage ferritin nanocages are found in all kingdoms and are essential in eukaryotes; they play a central role in iron homeostasis, where they protect the cell from toxic free $Fe^{2+}$ by oxidizing it and storing the resulting $Fe^{3+}$ as ferrihydrite minerals within their central cavity.

The encapsulin-associated enzymes are sequestered within the icosahedral shell through interactions between the shell's inner surface and a short localization sequence (Gly-Ser-Leu-Lys) appended to their C-termini (*Sutter et al., 2008*). This motif is well-conserved, and the addition of this sequence to heterologous proteins is sufficient to direct them to the interior of encapsulins (*Rurup et al., 2014*; *2015*; *Cassidy-Amstutz et al., 2016*).

A recent study of the *Myxococcus xanthus* encapsulin showed that it sequesters a number of different EncFtn proteins and acts as an 'iron-megastore' to protect these bacteria from oxidative stress (*McHugh et al., 2014*). At 32 nm in diameter, it is much larger than other members of the ferritin superfamily, such as the 12 nm 24-subunit classical ferritin nanocage and the 8 nm 12-subunit Dps (DNA-binding protein from starved cells) complex (*Grant et al., 1998*; *Andrews, 2010*); and is thus capable of sequestering up to ten times more iron than these ferritins (*McHugh et al., 2014*). The primary sequences of EncFtn proteins have Glu-X-X-His metal coordination sites, which are shared features of the ferritin family proteins (*Andrews, 2010*). Secondary structure prediction identifies two major $\alpha$-helical regions in these proteins; this is in contrast to other members of the ferritin superfamily, which have four major $\alpha$-helices (*Supplementary file 1*). The 'half-ferritin' primary sequence of the EncFtn family and their association with encapsulin nanocompartments suggests a distinct biochemical and structural organization to other ferritin family proteins. The *Rhodospirillum rubrum* EncFtn protein (Rru_A0973) shares 33% protein sequence identity with the *M. xanthus* (MXAN_4464), 53% with the *T. maritima* (Tmari_0787), and 29% with the *P. furiosus*

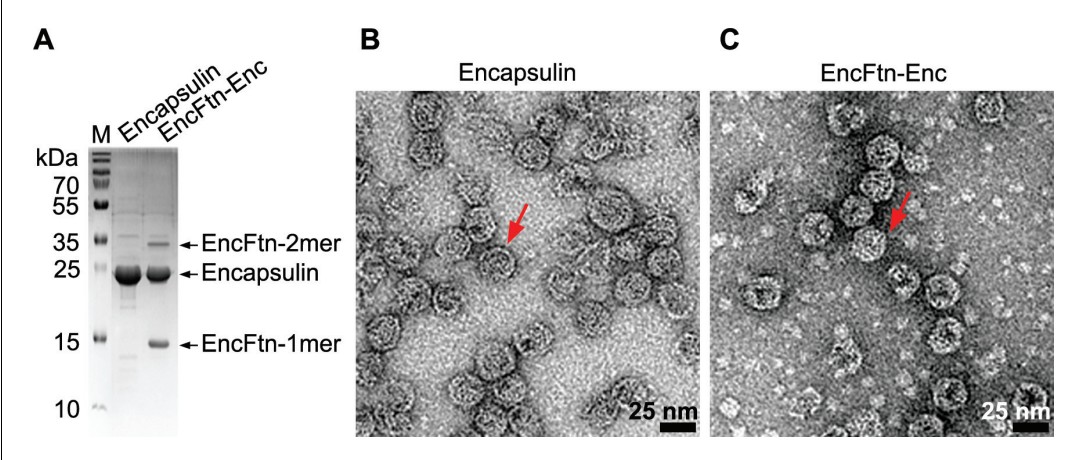

**Figure 1.** Purification of recombinant *R. rubrum* encapsulin nanocompartments. (**A**) Recombinantly expressed encapsulin (Enc) and co-expressed EncFtn-Enc were purified by sucrose gradient ultracentrifugation from *E. coli* B834(DE3) grown in SeMet medium. Samples were resolved by 18% acrylamide SDS-PAGE; the position of the proteins found in the complexes as resolved on the gel are shown with arrows. (**B/C**) Negative stain TEM image of recombinant encapsulin and EncFtn-Enc nanocompartments. Samples were imaged at 143,000 x magnification, with scale bar shown as 25 nm. Representative encapsulin and EncFtn-Enc complexes are indicated with red arrows.

The following figure supplement is available for figure 1:

**Figure supplement 1.** Full-frame transmission electron micrographs of *R. rubrum* nanocompartments.

(PF1192) homologues. The GXXH motifs are strictly conserved in each of these species (*Supplementary file 1*).

Here we investigate the structure and biochemistry of EncFtn in order to understand iron storage within the encapsulin nanocompartment. We have produced recombinant encapsulin (Enc) and EncFtn from the aquatic purple-sulfur bacterium *R. rubrum*, which serves as a model organism for the study of the control of the bacterial nitrogen fixation machinery (*Pope et al., 1985*), in *Escherichia coli*. Analysis by transmission electron microscopy (TEM) indicates that their co-expression leads to the production of an icosahedral nanocompartment with encapsulated EncFtn. The crystal structure of a truncated hexahistidine-tagged variant of the EncFtn protein (EncFtn$_{sH}$) shows that it forms a decameric structure with an annular 'ring-doughnut' topology, which is distinct from the four-helical bundles of the 24meric ferritins (*Lawson et al., 1991*) and dodecahedral DPS proteins (*Grant et al., 1998*). We identify a symmetrical iron bound ferroxidase center (FOC) formed between subunits in the decamer and additional metal-binding sites close to the center of the ring and on the outer surface. We also demonstrate the metal-dependent assembly of EncFtn decamers using native PAGE, analytical gel-filtration, and native mass spectrometry. Biochemical assays show that EncFtn is active as a ferroxidase enzyme. Through site-directed mutagenesis we show that the conserved glutamic acid and histidine residues in the FOC influence protein assembly and activity. We use our combined structural and biochemical data to propose a model for the EncFtn-catalyzed sequestration of iron within the encapsulin shell.

## Results

### Assembly of *R. rubrum* EncFtn encapsulin nanocompartments in E. coli

We produced recombinant *R. rubrum* encapsulin nanocompartments in *E. coli* by co-expression of the encapsulin (Rru_A0974) and EncFtn (Rru_A0973) proteins, and purified these by sucrose gradient ultra-centrifugation (*Figure 1A*) (*Sutter et al., 2008*). TEM imaging of uranyl acetate-stained samples revealed that, when expressed in isolation, the encapsulin protein forms empty compartments with an average diameter of 24 nm (*Figure 1B* and *Figure 1—figure supplement 1A/C*), consistent with the appearance and size of the *T. maritima* encapsulin (*Sutter et al., 2008*). We were not able

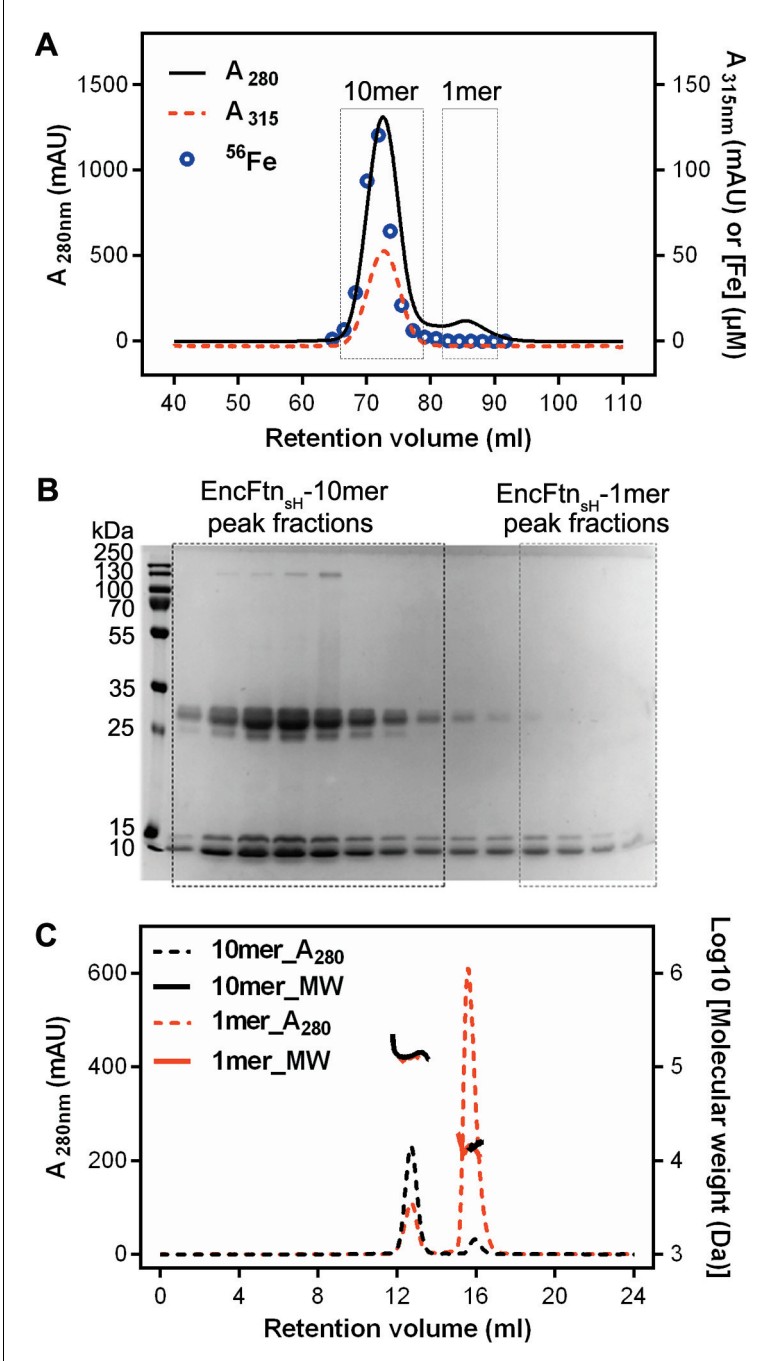

**Figure 2.** Purification of recombinant *R. rubrum* EncFtn$_{sH}$. (**A**) Recombinant SeMet-labeled EncFtn$_{sH}$ produced with 1 mM Fe(NH$_4$)$_2$(SO$_4$)$_2$ in the growth medium was purified by nickel affinity chromatography and size-exclusion chromatography using a Superdex 200 16/60 column (GE Healthcare). Chromatogram traces measured at 280 nm and 315 nm are shown with the results from ICP-MS analysis of the iron content of the fractions collected during the experiment. The peak around 73 ml corresponds to a molecular weight of around 130 kDa when compared to calibration standards; this is consistent with a decamer of EncFtn$_{sH}$. The small peak at 85 ml corresponds to the 13 kDa monomer compared to the standards. Only the decamer peak contains significant amounts of iron as indicated by the ICP-MS analysis. (**B**) Peak fractions from the gel filtration run were resolved by 15% acrylamide SDS-PAGE and stained with Coomassie blue stain. The bands around 13 kDa and 26 kDa correspond to EncFtn$_{sH}$, as identified by MALDI peptide mass fingerprinting. The band at 13 kDa is consistent with the monomer mass, while the band at 26 kDa is consistent with a dimer of EncFtn$_{sH}$. The dimer species only appears in the decamer fractions. (**C**) SEC-MALLS analysis of EncFtn$_{sH}$ from decamer fractions and monomer fractions allows assignment of

*Figure 2 continued on next page*

*Figure 2 continued*
an average mass of 132 kDa to decamer fractions and 13 kDa to monomer fractions, consistent with decamer and monomer species (*Table 2*).

to resolve any higher-order structures of EncFtn by TEM. Protein purified from co-expression of the encapsulin and EncFtn resulted in 24 nm compartments with regions in the center that exclude stain, consistent with the presence of the EncFtn within the encapsulin shell (*Figure 1C* and *Figure 1—figure supplement 1B/C*).

## *R. rubrum* EncFtn forms a metal-ion stabilized decamer in solution

We purified recombinant *R. rubrum* EncFtn as both the full-length sequence (140 amino acids) and a truncated C-terminal hexahistidine-tagged variant (amino acids 1–96 plus the tag; herein EncFtn$_{sH}$). In both cases the elution profile from size-exclusion chromatography (SEC) displayed two peaks (*Figure 2A*). SDS-PAGE analysis of fractions from these peaks showed that the high molecular weight peak was partially resistant to SDS and heat-induced denaturation; in contrast, the low molecular weight peak was consistent with monomeric mass of 13 kDa (*Figure 2B*). MALDI peptide mass fingerprinting of these bands confirmed the identity of both as EncFtn. Inductively coupled plasma mass spectrometry (ICP-MS) analysis of the SEC fractions showed 100 times more iron in the oligomeric fraction than the monomer (*Figure 2A*, blue scatter points; *Table 1*), suggesting that EncFtn oligomerization is associated with iron binding. In order to determine the iron-loading stoichiometry in the EncFtn complex, further ICP-MS experiments were performed using selenomethionine (SeMet)-labelled protein EncFtn (*Table 1*). In these experiments, we observed sub-stoichiometric metal binding, which is in contrast to the classical ferritins (*Le Brun et al., 2010*). Size-exclusion chromatography with multi-angle laser light scattering (SEC-MALLS) analysis of

**Table 1.** Determination of the Fe/EncFtn$_{sH}$ protein ratio by ICP-MS. EncFtn$_{sH}$ was purified as a SeMet derivative from *E. coli* B834(DE3) cells grown in SeMet medium with 1 mM Fe(NH$_4$)$_2$(SO$_4$)$_2$. Fractions from SEC were collected, acidified and analysed by ICP-MS. EncFtn$_{sH}$ concentration was calculated based on the presence of two SeMet per mature monomer. Samples where the element was undetectable are labelled with n.d. These data were collected from EncFtn$_{sH}$ fractions from a single gel-filtration run.

| Peak | EncFtn$_{sH}$ retention volume (ml) | Element concentration (µM) | | | | Derived EncFtn$_{sH}$ concentration (µM) | Derived Fe/ EncFtn$_{sH}$ monomer |
|---|---|---|---|---|---|---|---|
| | | Ca | Fe | Zn | Se | | |
| Decamer | 66.5 | n.d. | 6.7 | n.d. | 24.6 | 12.3 | 0.5 |
| | 68.3 | n.d. | 28.4 | n.d | 124.5 | 62.3 | 0.5 |
| | 70.1 | 2.9 | 93.7 | 2.4 | 301.7 | 150.9 | 0.6 |
| | 71.9 | 6.9 | 120.6 | 3.7 | 379.8 | 189.9 | 0.6 |
| | 73.7 | 1.9 | 64.4 | 0.8 | 240.6 | 120.3 | 0.5 |
| | 75.5 | 0.9 | 21.1 | n.d. | 101.7 | 50.8 | 0.4 |
| | 77.3 | n.d. | 6.2 | n.d. | 42.6 | 21.3 | 0.3 |
| | 79.1 | 0.1 | 2.4 | n.d. | 26.5 | 13.3 | 0.2 |
| | 80.9 | 1.0 | 1.5 | n.d. | 22.3 | 11.2 | 0.1 |
| | 82.7 | n.d. | 0.2 | n.d. | 29.2 | 14.6 | n.d |
| Monomer | 84.5 | n.d. | 0.1 | n.d. | 34.9 | 17.5 | n.d |
| | 86.3 | n.d. | n.d | n.d. | 28.9 | 14.4 | n.d |
| | 88.1 | n.d. | n.d. | n.d. | 17.4 | 8.7 | n.d. |
| | 89.9 | n.d. | n.d. | n.d. | 5.5 | 2.8 | n.d. |
| | 91.7 | n.d. | n.d. | n.d. | 0.1 | 0.07 | 0.2 |

**Table 2.** Estimates of EncFtn$_{sH}$ molecular weight from SEC-MALLS analysis. EncFtn$_{sH}$ was purified from *E. coli* BL21(DE3) grown in minimal medium (MM) by nickel affinity chromatography and size-exclusion chromatography. Fractions from two peaks (decamer and monomer) were pooled separately (*Figure 1C*) and analysed by SEC-MALLS using a Superdex 200 10/300 GL column (GE Healthcare) and Viscotek SEC-MALLS instruments (Malvern Instruments) (*Figure 2C*). The decamer and monomer peaks were both symmetric and monodisperse, allowing the estimation of the molecular weight of the species in these fractions (*Folta-Stogniew, 2006*). The molecular weights are quoted to the nearest kDa due to the resolution limit of the instrument. The proteins analyzed by SEC-MALLS came from single protein preparation.

| Molecular Weight (kDa) | Decamer peak | Monomer peak |
|---|---|---|
| Theoretical | 133 | 13 |
| EncFtn$_{sH}$-decamer fractions | 132 | 15 |
| EncFtn$_{sH}$-monomer fractions | 126 | 13 |

samples taken from each peak gave calculated molecular weights consistent with a decamer for the high molecular weight peak and a monomer for the low molecular weight peak (*Figure 2C*, *Table 2*).

We purified EncFtn$_{sH}$ from *E. coli* grown in MM with or without the addition of 1 mM Fe(NH$_4$)$_2$(SO$_4$)$_2$. The decamer to monomer ratio in the sample purified from cells grown in iron-supplemented media was 4.5, while that from the iron-free media was 0.11, suggesting that iron induces the oligomerization of EncFtn$_{sH}$ *in vivo* (*Figure 3A*, *Table 3*). To test the metal-dependent oligomerization of EncFtn$_{sH}$ *in vitro*, we incubated the protein with various metal cations and subjected samples to analytical SEC and non-denaturing PAGE. Of the metals tested, only Fe$^{2+}$, Zn$^{2+}$ and Co$^{2+}$ induced the formation of significant amounts of the decamer (*Figure 3B*, *Figure 3—figure supplement 1/2*). While Fe$^{2+}$ induces the multimerization of EncFtn$_{sH}$, Fe$^{3+}$ in the form of FeCl$_3$ does not have this effect on the protein, highlighting the apparent preference this protein has for the ferrous form of iron. To determine if the oligomerization of EncFtn$_{sH}$ was concentration dependent we performed analytical SEC at 90 and 700 µM protein concentration (*Figure 3C*). At the higher concentration, no increase in the decameric form of EncFtn was observed; however, the shift in the major peak from the position of the monomer species indicated a tendency to dimerize at high concentration.

## Crystal structure of EncFtn$_{sH}$

We determined the crystal structure of EncFtn$_{sH}$ by molecular replacement to 2.0 Å resolution (see *Table 1* for X-ray data collection and refinement statistics). The crystallographic asymmetric unit contained thirty monomers of EncFtn with visible electron density for residues 7 – 96 in each chain. The protein chains were arranged as three identical annular decamers, each with D5 symmetry. The decamer has a diameter of 7 nm and thickness of 4 nm (*Figure 4A*). The monomer of EncFtn has an N-terminal 3$_{10}$-helix that precedes two 4 nm long antiparallel α-helices arranged with their long axes at 25° to each other; these helices are followed by a shorter 1.4 nm helix projecting at 70° from α2 (*Figure 4B*). The C-terminal region of the crystallized construct extends from the outer circumference of the ring, indicating that the encapsulin localization sequence in the full-length protein is on the exterior of the ring and is thus free to interact with its binding site on the encapsulin shell protein (*Sutter et al., 2008*).

The monomer of EncFtn$_{sH}$ forms two distinct dimer interfaces within the decamer (*Figure 4 C/D*). The first dimer is formed from two monomers arranged antiparallel to each other, with α1 from each monomer interacting along their lengths and α3 interdigitating with α2 and α3 of the partner chain. This interface buries one third of the surface area from each partner and is stabilized by thirty hydrogen bonds and fourteen salt bridges (*Figure 4C*). The second dimer interface forms an antiparallel four-helix bundle between helices 1 and 2 from each monomer (*Figure 4D*). This interface is less extensive than the first and is stabilized by twenty-one hydrogen bonds, six salt bridges, and a number of metal ions.

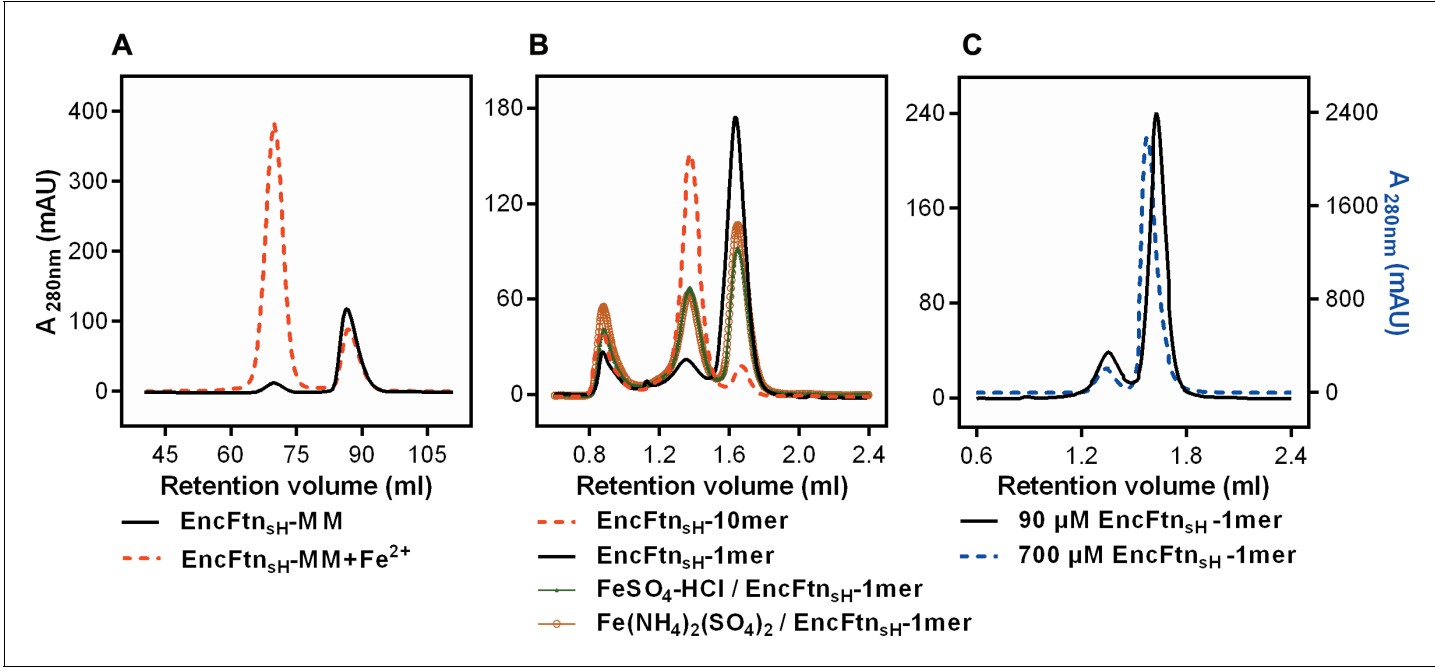

**Figure 3.** Effect of $Fe^{2+}$ and protein concentration on the oligomeric state of EncFtn$_{sH}$ in solution. (A) Recombinant EncFtn$_{sH}$ was purified by Gel filtration Superdex 200 chromatography from *E. coli* BL21(DE3) grown in MM or in MM supplemented with 1 mM $Fe(NH_4)_2(SO_4)_2$ (MM+$Fe^{2+}$). A higher proportion of decamer (peak between 65 and 75 ml) is seen in the sample purified from MM+$Fe^{2+}$ compared to EncFtn$_{sH}$-MM, indicating that $Fe^{2+}$ facilitates the multimerization of EncFtn$_{sH}$ *in vivo*. (B) EncFtn$_{sH}$-monomer was incubated with one molar equivalent of $Fe^{2+}$ salts for two hours prior to analytical gel-filtration using a Superdex 200 PC 3.2/30 column (GE Healthcare). Both $Fe^{2+}$ salts tested induced the formation of decamer indicated by the peak between 1.2 and 1.6 ml. Monomeric and decameric samples of EncFtn$_{sH}$ are shown as controls. Peaks around 0.8 ml were seen as protein aggregation. (C) Analytical gel filtration of EncFtn monomer at different concentrations to illustrate the effect of protein concentration on multimerization. The major peak shows a shift towards a dimer species at high concentration of protein, but the ratio of this peak (1.5–1.8 ml) to the decamer peak (1.2–1.5 ml) does not change when compared to the low concentration sample.

The following figure supplements are available for figure 3:

**Figure supplement 1.** Effect of metal ions on the oligomeric state of EncFtn$_{sH}$ in solution.

**Figure supplement 2.** PAGE analysis of the effect of metal ions on the oligomeric state of EncFtn$_{sH}$.

The arrangement of ten monomers in alternating orientation forms the decamer of EncFtn, which assembles as a pentamer of dimers (*Figure 4A*). Each monomer lies at 45° relative to the vertical central-axis of the ring, with the N-termini of alternating subunits capping the center of the ring at each end, while the C-termini are arranged around the circumference. The central hole in the ring is 2.5 nm at its widest in the center of the complex, and 1.5 nm at its narrowest point near the outer surface, although it should be noted that a number of residues at the N-terminus are not visible in the crystallographic electron density and these may occupy the central channel. The surface of the decamer has distinct negatively charged patches, both within the central hole and on the outer circumference, which form spokes through the radius of the complex (*Figure 4—figure supplement 1*).

## EncFtn ferroxidase center

The electron density maps of the initial EncFtn$_{sH}$ model displayed significant positive peaks in the mFo-DFc map at the center of the 4-helix bundle dimer (*Figure 5—figure supplement 1*). Informed by the ICP-MS data indicating the presence of iron in the protein we collected diffraction data at the experimentally determined iron absorption edge (1.74 Å) and calculated an anomalous difference Fourier map using this data. Inspection of this map showed two 10-sigma peaks between residues Glu32, Glu62 and His65 of two adjacent chains, and a statistically smaller 5-sigma peak between

**Table 3.** Gel-filtration peak area ratios for EncFtn$_{sH}$ decamer and monomer on addition of different metal ions. EncFtn$_{sH}$ was produced in *E. coli* BL21(DE3) cultured in MM and MM with 1 mM Fe(NH$_4$)$_2$(SO$_4$)$_2$ (MM+Fe$^{2+}$) and purified by gel-filtration chromatography using an Superdex 200 16/60 column (GE Healthcare). Monomer fractions of EncFtn$_{sH}$ purified from MM were pooled and run in subsequent analytical gel-filtration runs over the course of three days. Samples of EncFtn$_{sH}$ monomer were incubated with one molar equivalent of metal ion salts at room temperature for two hours before analysis by analytical gel filtration chromatography (AGF) using a Superdex 200 10/300 GL column. The area for resulting protein peaks were calculated using the Unicorn software (GE Healthcare); peak ratios were calculated to quantify the propensity of EncFtn$_{sH}$ to multimerize in the presence of the different metal ions. The change in the ratios of monomer to decamer over the three days of experiments may be a consequence of experimental variability, or the propensity of this protein to equilibrate towards decamer over time. The increased decamer: monomer ratio seen in the presence of Fe$^{2+}$, Co$^{2+}$, and Zn$^{2+}$ indicates that these metal ions facilitate multimerization of the EncFtn$_{sH}$ protein, while the other metal ions tested do not appear to induce multimerization. The analytical gel filtration experiment was repeated twice using two independent preparations of protein, of which values calculated from one sample are presented here.

| Method | Sample | Monomer area | Decamer area | Decamer/Monomer |
|---|---|---|---|---|
| Gel filtration Superdex 200 chromatography | EncFtn$_{sH}$-MM | 64.3 | 583.6 | 0.1 |
| | EncFtn$_{sH}$-MM+Fe$^{2+}$ | 1938.4 | 426.4 | 4.5 |
| Analytical Gel filtration Day1 | EncFtn$_{sH}$-decamer fractions | 20.2 | 1.8 | 11.2 |
| | EncFtn$_{sH}$-monomer fractions | 2.9 | 21.9 | 0.1 |
| | Fe(NH$_4$)$_2$(SO$_4$)$_2$/EncFtn$_{sH}$-monomer | 11.0 | 13.0 | 0.8 |
| | FeSO$_4$-HCl/EncFtn$_{sH}$-monomer | 11.3 | 11.4 | 1.0 |
| Analytical Gel filtration Day2 | EncFtn$_{sH}$-monomer fractions | 8.3 | 22.8 | 0.4 |
| | CoCl$_2$/EncFtn$_{sH}$-monomer | 17.7 | 14.5 | 1.2 |
| | MnCl$_2$/EncFtn$_{sH}$-monomer | 3.1 | 30.5 | 0.1 |
| | ZnSO$_4$/EncFtn$_{sH}$-monomer | 20.4 | 9.0 | 2.3 |
| | FeCl$_3$/EncFtn$_{sH}$-monomer | 3.9 | 28.6 | 0.1 |
| Analytical Gel filtration Day3 | EncFtn$_{sH}$-monomer fractions | 6.3 | 23.4 | 0.3 |
| | MgSO$_4$/EncFtn$_{sH}$-monomer | 5.8 | 30.2 | 0.2 |
| | Ca acetate/EncFtn$_{sH}$-monomer | 5.6 | 25.2 | 0.2 |

residues Glu31 and Glu34 of the two chains. Modeling metal ions into these peaks and refinement of the anomalous scattering parameters allowed us to identify these as two iron ions and a calcium ion respectively (*Figure 5A*). An additional region of asymmetric electron density near the di-iron binding site in the mFo-DFc map was modeled as glycolic acid, presumably a breakdown product of the PEG 3350 used for crystallization. This di-iron center has an Fe-Fe distance of 3.5 Å, Fe-Glu-O distances between 2.3 and 2.5 Å, and Fe-His-N distances of 2.5 Å (*Figure 5B*). This coordination geometry is consistent with the di-nuclear ferroxidase center (FOC) found in ferritin (*Bertini et al., 2012*). It is interesting to note that although we did not add any additional iron to the crystallization trials, the FOC was fully occupied with iron in the final structure, implying that this site has a very high affinity for iron.

The calcium ion coordinated by Glu31 and Glu34 adopts heptacoordinate geometry, with coordination distances of 2.5 Å between the metal ion and carboxylate oxygens of Glu31 and Glu34 (E31/34-site). A number of ordered solvent molecules are also coordinated to this metal ion at a distance of 2.5 Å. This heptacoordinate geometry is common in crystal structures with calcium ions (*Figure 5C*) (*Katz et al., 1996*). While ICP-MS indicated that there were negligible amounts of calcium in the purified protein, the presence of 140 mM calcium acetate in the crystallization mother liquor favors the coordination of calcium at this site. The fact that the protein does not multimerize in solution in the presence of Fe$^{3+}$ may indicate that these metal binding sites have a lower affinity for the ferric form of iron, which is the product of the ferroxidase reaction. A number of additional metal-ions were present at the outer circumference of at least one decamer in the asymmetric unit (*Figure 5D*). These ions are coordinated by His57, Glu61 and Glu64 from both chains in the FOC

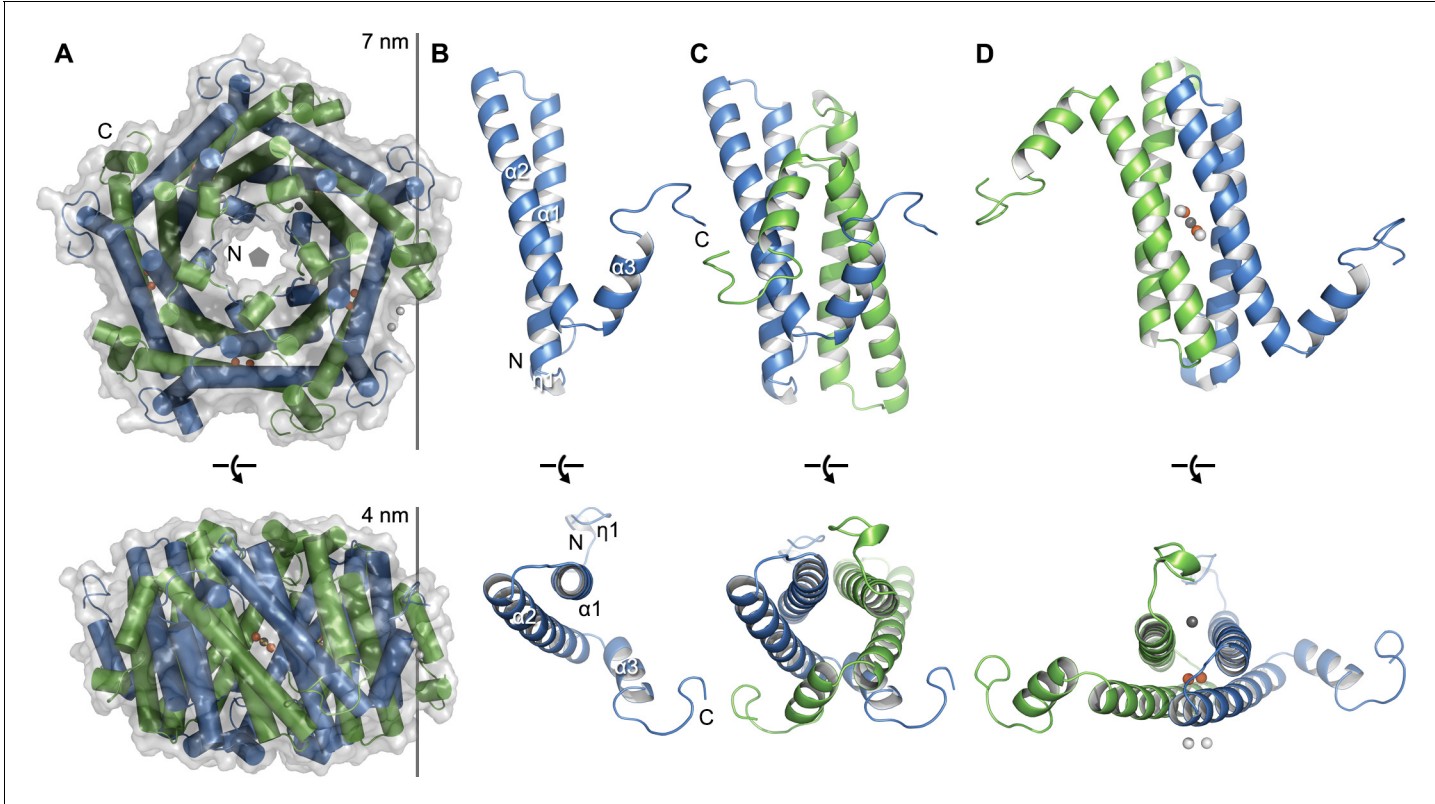

**Figure 4.** Crystal structure of EncFtn$_{sH}$. (A) Overall architecture of EncFtn$_{sH}$. Transparent solvent accessible surface view with α-helices shown as tubes and bound metal ions as spheres. Alternating subunits are colored blue and green for clarity. The doughnut-like decamer is 7 nm in diameter and 4.5 nm thick. (B) Monomer of EncFtn$_{sH}$ shown as a secondary structure cartoon. (C/D) Dimer interfaces formed in the decameric ring of EncFtn$_{sH}$. Subunits are shown as secondary structure cartoons and colored blue and green for clarity. Bound metal ions are shown as orange spheres for $Fe^{3+}$ and grey and white spheres for $Ca^{2+}$.

The following figure supplement is available for figure 4:

**Figure supplement 1.** Electrostatic surface of EncFtn$_{sH}$.

dimer and are 4.5 Å apart; Fe-Glu-O distances are between 2.5 and 3.5 Å and the Fe-His-N distances are 4 and 4.5 Å.

Structural alignment of the di-iron binding site of EncFtn$_{sH}$ to the FOC of *Pseudo-nitzschia multis-eries* ferritin (PmFtn, PDB ID: 4ITW) reveals a striking similarity between the metal binding sites of EncFtn$_{sH}$ and the classical ferritins (*Pfaffen et al., 2013*) (*Figure 6A*). The di-iron site of EncFtn$_{sH}$ is by necessity symmetrical, as it is formed through a dimer interface, while the FOC of ferritin does not have these constraints and varies in different species at a position equivalent to His65 of the second EncFtn monomer in the FOC interface (His65') (*Figure 6A*). Structural superimposition of the FOCs of ferritin and EncFtn brings the four-helix bundle of the ferritin fold into close alignment with the EncFtn dimer, showing that the two families of proteins have essentially the same architecture around the di-iron center (*Figure 6B*). The linker connecting helices 2 and 3 of ferritin is congruent with the start of the C-terminal helix of one EncFtn monomer and the N-terminal $3_{10}$ helix of the second monomer (*Figure 6C*).

## Mass spectrometry of the EncFtn assembly

In order to confirm the assignment of the oligomeric state of EncFtn$_{sH}$ and investigate further the $Fe^{2+}$-dependent assembly, we used native nano-electrospray ionization (nESI) and ion-mobility mass spectrometry (IM-MS). As described above, by recombinant production of EncFtn$_{sH}$ in minimal media we were able to limit the bioavailability of iron. Native MS analysis of EncFtn$_{sH}$ produced in

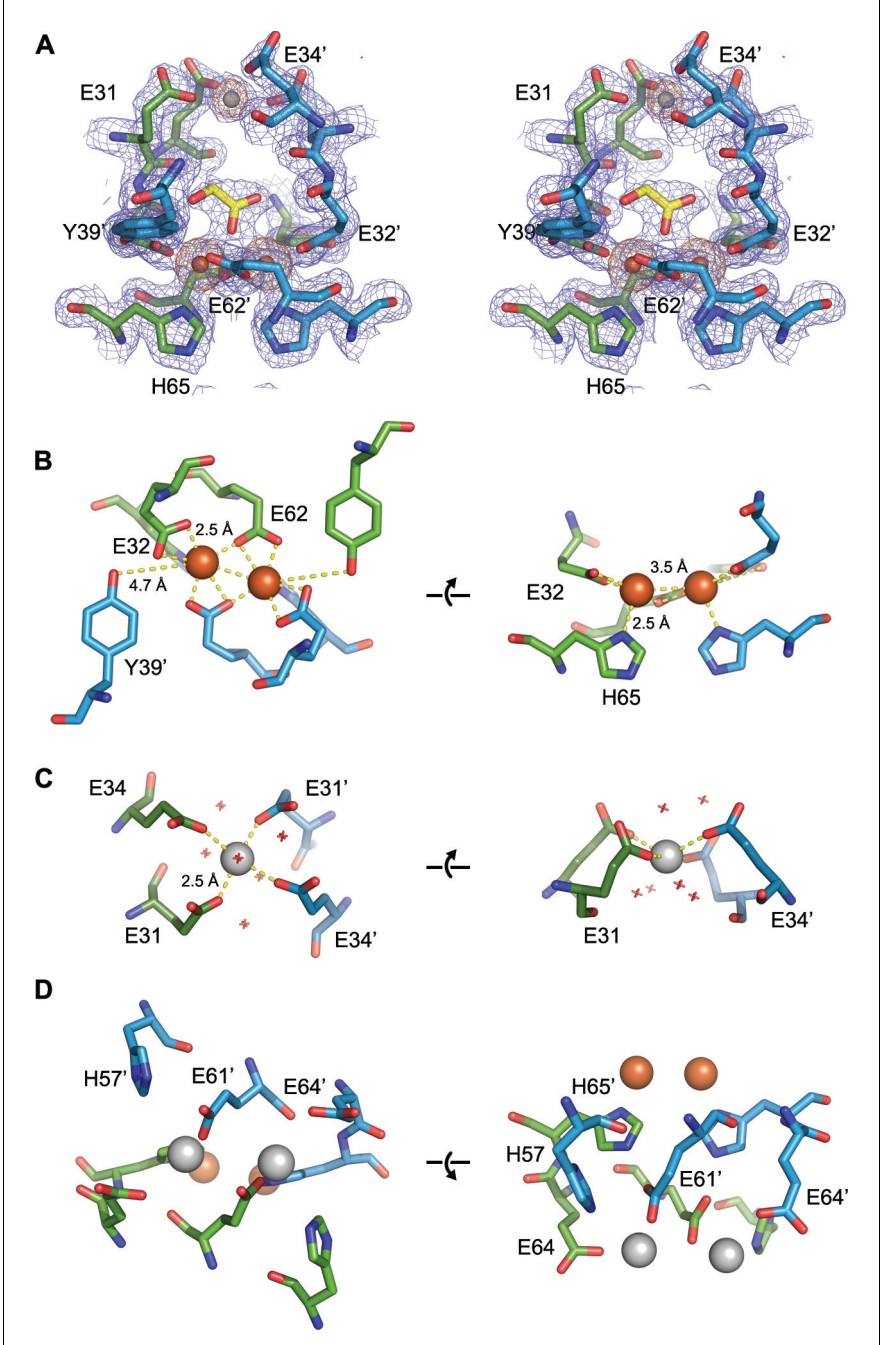

**Figure 5.** EncFtn_sH metal binding sites. (**A**) Wall-eyed stereo view of the metal-binding dimerization interface of EncFtn_sH. Protein residues are shown as sticks with blue and green carbons for the different subunits, iron ions are shown as orange spheres and calcium as grey spheres, and the glycolic acid ligand is shown with yellow carbon atoms coordinated above the di-iron center. The 2mFo-DFc electron density map is shown as a blue mesh contoured at 1.5 σ and the NCS-averaged anomalous difference map is shown as an orange mesh and contoured at 10 σ. (**B**) Iron coordination within the FOC including residues Glu32, Glu62, His65 and Tyr39 from two chains. Protein and metal ions are shown as in **A**. Coordination between the protein and iron ions is shown as yellow dashed lines with distances indicated. (**C**) Coordination of calcium within the dimer interface by four glutamic acid residues (E31 and E34 from two chains). The calcium ion is shown as a grey sphere and water molecules involved in the coordination of the calcium ion are shown as crosses. (**D**) Metal coordination site on the outer surface of EncFtn_sH. The two calcium ions are coordinated by residues His57, Glu61 and Glu64 from the two chains of the FOC dimer, and are located at the outer surface of the complex, positioned 10 Å away from the FOC iron.

*Figure 5 continued on next page*

*Figure 5 continued*

The following figure supplement is available for figure 5:

**Figure supplement 1.** Putative ligand-binding site in EncFtn$_{sH}$.

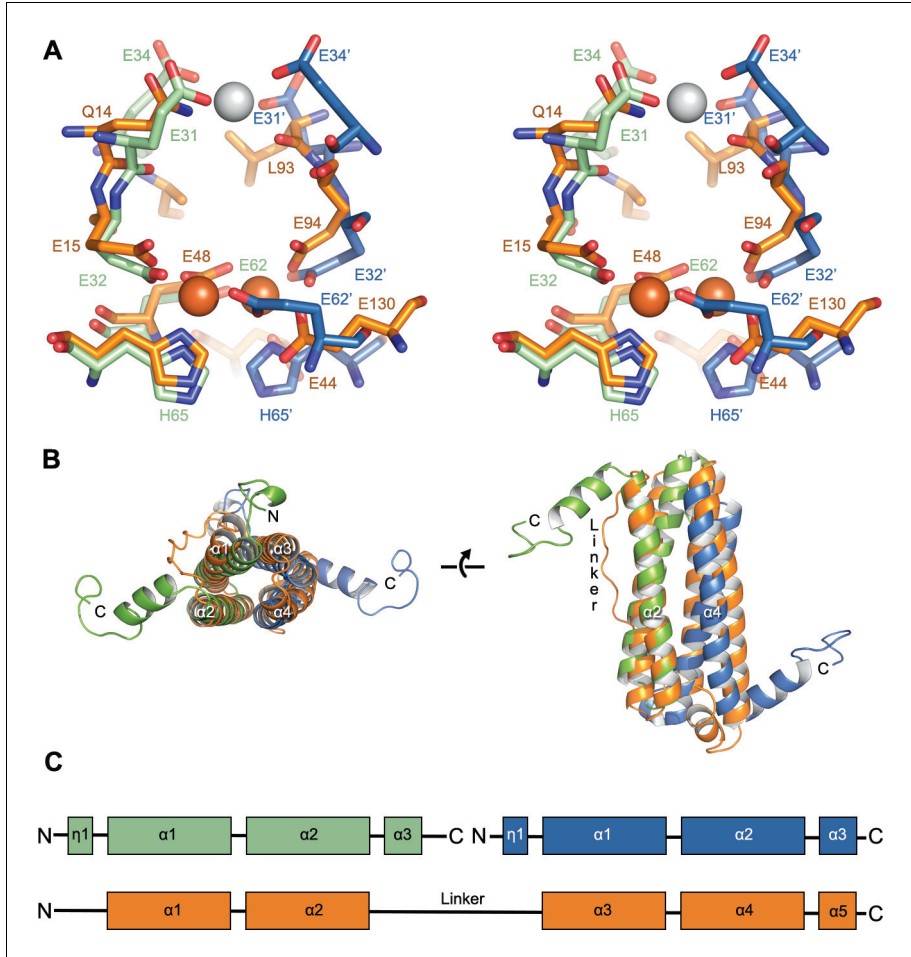

**Figure 6.** Comparison of the symmetric metal ion binding site of EncFtn$_{sH}$ and the ferritin FOC. (**A**) Structural alignment of the FOC residues in a dimer of EncFtn$_{sH}$ (green/blue) with a monomer of *Pseudo-nitzschia multiseries* ferritin (PmFtn) (PDBID: 4ITW) (orange) (*Pfaffen et al., 2013*). Iron ions are shown as orange spheres and a single calcium ion as a grey sphere. Residues within the FOC are conserved between EncFtn and ferritin PmFtn, with the exception of residues in the position equivalent to H65' in the second subunit in the dimer (blue). The site in EncFtn with bound calcium is not present in other family members. (**B**) Secondary structure of aligned dimeric EncFtn$_{sH}$ and monomeric ferritin highlighting the conserved four-helix bundle. EncFtn$_{sH}$ monomers are shown in green and blue and aligned PmFtn monomer in orange as in **A**. (**C**) Cartoon of secondary structure elements in EncFtn dimer and ferritin. In the dimer of EncFtn that forms the FOC, the C-terminus of the first monomer (green) and N-terminus of the second monomer (blue) correspond to the position of the long linker between α2 and α3 in ferritin PmFtn.

The following figure supplement is available for figure 6:

**Figure supplement 1.** Comparison of quaternary structure of EncFtn$_{sH}$ and ferritin.

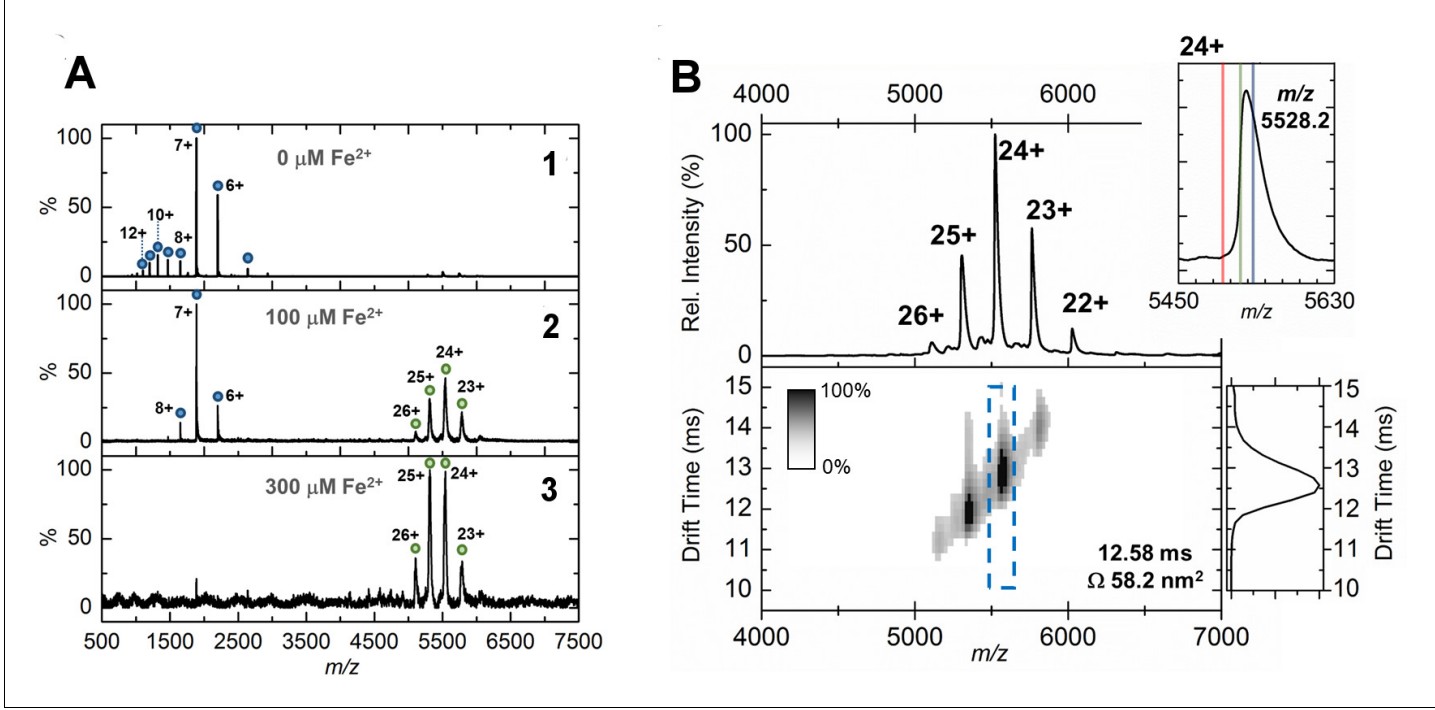

**Figure 7.** Native mass spectrometry and ion mobility analysis of iron loading in EncFtn$_{sH}$. All spectra were acquired in 100 mM ammonium acetate, pH 8.0 with a protein concentration of 5 μM. (**A**) Native nanoelectrospray ionization (nESI) mass spectrometry of EncFtn$_{sH}$ at varying iron concentrations. A1, nESI spectrum of iron-free EncFtn$_{sH}$ displays a charge state distribution consistent with EncFtn$_{sH}$ monomer (blue circles, 13,194 Da). Addition of 100 μM (**A2**) and 300 μM (**A3**) Fe$^{2+}$ results in the appearance of a second higher molecular weight charge state distribution consistent with a decameric assembly of EncFtn$_{sH}$ (green circles, 132.6 kDa). (**B**) Ion mobility (IM)-MS of the iron-bound holo-EncFtn$_{sH}$ decamer. *Top*, Peaks corresponding to the 22 + to 26+ charge states of a homo-decameric assembly of EncFtn$_{sH}$ are observed (132.6 kDa). *Top Insert*, Analysis of the 24+ charge state of the assembly at *m/z* 5528.2 Th. The theoretical average *m/z* of the 24+ charge state with no additional metals bound is marked by a red line (5498.7 Th); the observed *m/z* of the 24+ charge state indicates that the EncFtn$_{sH}$ assembly binds between 10 (green line, 5521.1 Th) and 15 Fe ions (blue line, 5532.4 Th) per decamer. *Bottom*, The arrival time distributions (ion mobility data) of all ions in the EncFtn$_{sH}$ charge state distribution displayed as a greyscale heat map (linear intensity scale). *Bottom right*, The arrival time distribution of the 24+ charge state (dashed blue box) has been extracted and plotted. The drift time for this ion is shown (ms), along with the calibrated collision cross section (CCS), Ω (nm$^2$).

The following figure supplements are available for figure 7:

**Figure supplement 1.** Native IM-MS analysis of the apo-EncFtn$_{sH}$ monomer.

**Figure supplement 2.** Gas-phase disassembly of the holo-EncFtn$_{sH}$ decameric assembly.

this way displayed a charge state distribution consistent with an EncFtn$_{sH}$ monomer (blue circles, *Figure 7A1*) with an average neutral mass of 13,194 Da, in agreement with the predicted mass of the EncFtn$_{sH}$ protein (13,194.53 Da). Under these conditions, no significant higher order assembly was observed and the protein did not have any coordinated metal ions. Titration with Fe$^{2+}$ directly before native MS analysis resulted in the appearance of a new charge state distribution, consistent with an EncFtn$_{sH}$ decameric assembly (+22 to +26; 132.65 kDa) (*Figure 7A2/3*). After instrument optimization, the mass resolving power achieved was sufficient to assign iron-loading in the complex to between 10 and 15 Fe ions per decamer (*Figure 7B*, inset top right), consistent with the presence of 10 irons in the FOC and the coordination of iron in the Glu31/34-site occupied by calcium in the crystal structure (Δmass observed ~0.67 kDa). MS analysis of EncFtn$_{sH}$ after addition of further Fe$^{2+}$ did not result in iron loading above this stoichiometry. Therefore, the extent of iron binding seen is limited to the FOC and Glu31/34 secondary metal binding site. These data suggest that the decameric assembly of EncFtn$_{sH}$ does not accrue iron in the same manner as classical ferritin, which is able to sequester around 4500 iron ions within its nanocage (*Mann et al., 1986*). Ion mobility

analysis of the EncFtn$_{sH}$ decameric assembly, collected with minimal collisional activation, suggested that it consists of a single conformation with a collision cross section (CCS) of 58.2 nm$^2$ (*Figure 7B*). This observation is in agreement with the calculated CCS of 58.7 nm$^2$ derived from our crystal structure of the EncFtn$_{sH}$ decamer (*Marklund, 2015*). By contrast, IM-MS measurements of the monomeric EncFtn$_{sH}$ at pH 8.0 under the same instrumental conditions revealed that the metal-free protein monomer exists in a wide range of charge states (+6 to +16) and adopts many conformations in the gas phase with collision cross sections ranging from 12 nm$^2$ to 26 nm$^2$ (*Figure 7—figure supplement 1*). These observations are indicative of an unstructured protein with little secondary or tertiary structure (*Beveridge et al., 2014*). Thus, IM-MS studies highlight that higher order structure in EncFtn$_{sH}$ is mediated/stabilized by metal binding, an observation that is in agreement with our solution studies. Taken together, these results suggest that di-iron binding, forming the FOC in EncFtn$_{sH}$, is required to stabilize the 4-helix bundle dimer interface, essentially reconstructing the classical ferritin-like fold; once stabilized, these dimers readily associate as pentamers, and the overall assembly adopts the decameric ring arrangement observed in the crystal structure.

We subsequently performed gas phase disassembly of the decameric EncFtn$_{sH}$ using collision-induced dissociation (CID) tandem mass spectrometry. Under the correct CID conditions, protein assemblies can dissociate with retention of subunit and ligand interactions, and thus provide structurally-informative evidence as to the topology of the original assembly; this has been termed 'atypical' dissociation (*Hall et al., 2013*). For EncFtn$_{sH}$, this atypical dissociation pathway was clearly

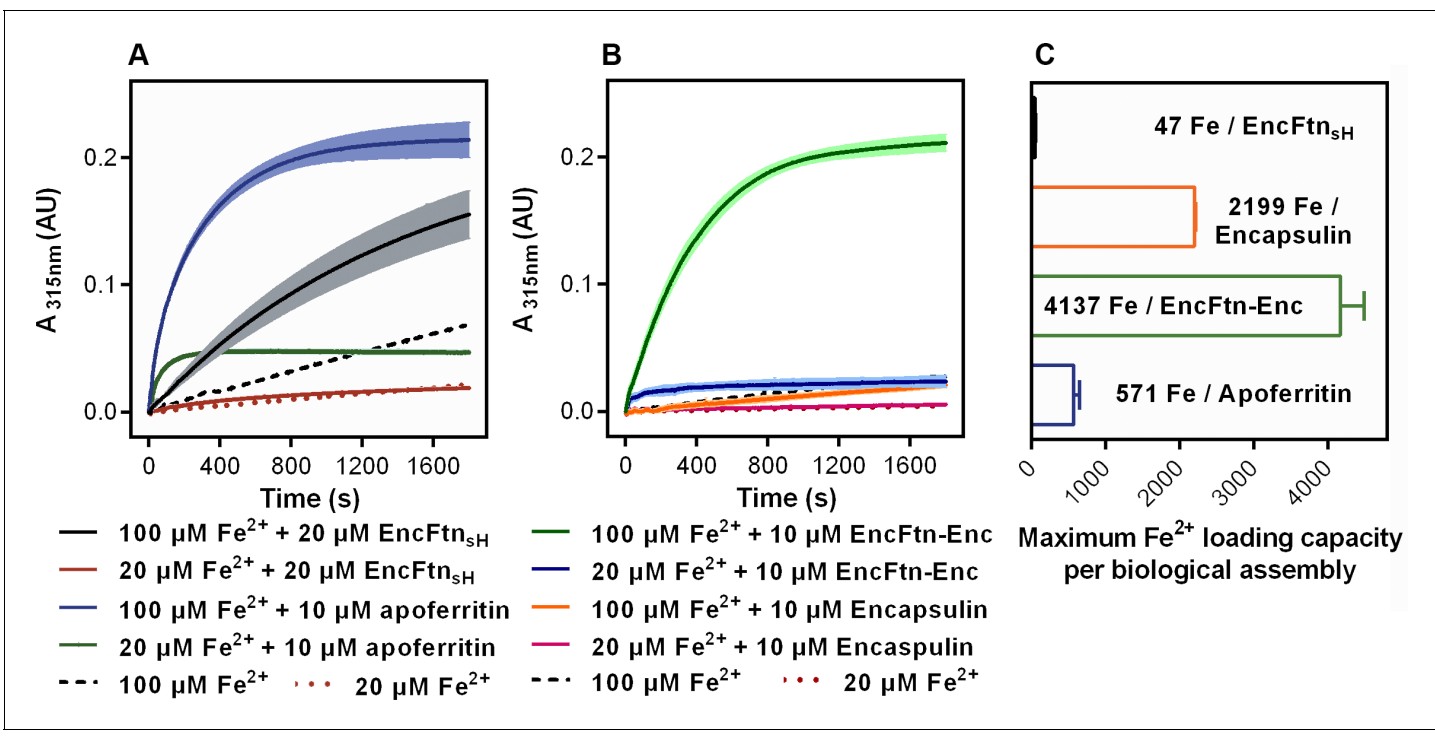

**Figure 8.** Spectroscopic evidence for the ferroxidase activity and comparison of iron loading capacity of apoferritin, EncFtn$_{sH}$, encapsulin, and EncFtn-Enc. (A) Apoferritin (10 μM monomer concentration) and EncFtn$_{sH}$ decamer fractions (20 μM monomer concentration, 10 μM FOC concentration) were incubated with 20 and 100 μM iron (2 and 10 times molar equivalent Fe$^{2+}$ per FOC) and progress curves of the oxidation of Fe$^{2+}$ to Fe$^{3+}$ at 315 nm were recorded in a spectrophotometer. The background oxidation of iron at 20 and 100 μM in enzyme-free controls are shown for reference. (B) Encapsulin and EncFtn-Enc complexes at 10 μM asymmetric unit concentration were incubated with Fe$^{2+}$ at 20 and 100 μM and progress curves for iron oxidation at A$_{315}$ were measured in a UV/visible spectrophotometer. Enzyme free controls for background oxidation of Fe$^{2+}$ are shown for reference. (C) Histogram of the iron loading capacity per biological assembly of EncFtn$_{sH}$, encapsulin, EncFtn-Enc and apoferritin. The results shown are for three technical replicates and represent the optimal iron loading by the complexes after three hours when incubated with Fe$^{2+}$.

The following figure supplement is available for figure 8:

**Figure supplement 1.** TEM visualization of iron-loaded bacterial nanocompartments and ferritin.

evident; CID of the EncFtn$_{sH}$ decamer resulted in the appearance of a dimeric EncFtn$_{sH}$ subcomplex containing 0, 1, or 2 iron ions (*Figure 7—figure supplement 2*). In light of the crystal structure, this observation can be rationalized as dissociation of the EncFtn$_{sH}$ decamer by disruption of the non-FOC interface with at least partial retention of the FOC interface and the FOC-Fe. Thus, this observation supports our crystallographic assignment of the overall topology of the EncFtn$_{sH}$ assembly as a pentameric assembly of dimers with two iron ions located at the FOC dimer interface. In addition, this analysis provides evidence that the overall architecture of the complex is consistent in the crystal, solution and gas phases.

## Ferroxidase activity

In light of the identification of an iron-loaded FOC in the crystal structure of EncFtn and our native mass spectrometry data, we performed ferroxidase and peroxidase assays to demonstrate the catalytic activity of this protein. In addition, we also assayed equine apoferritin, an example of a classical ferritin enzyme, as a positive control. Unlike the Dps family of ferritin-like proteins, EncFtn showed no peroxidase activity when assayed with the substrate *ortho*-phenylenediamine (*Pesek et al., 2011*). The ferroxidase activity of EncFtn$_{sH}$ was measured by recording the progress curve of $Fe^{2+}$ oxidation to $Fe^{3+}$ at 315 nm after addition of 20 and 100 µM $Fe^{2+}$ (2 and 10 times molar ratio $Fe^{2+}$/FOC). In both experiments the rate of oxidation was faster than background oxidation of $Fe^{2+}$ by molecular oxygen, and was highest for 100 µM $Fe^{2+}$ (*Figure 8A*). These data show that recombinant EncFtn$_{sH}$ acts as an active ferroxidase enzyme. When compared to apoferritin, EncFtn$_{sH}$ oxidized $Fe^{2+}$ at a slower rate and the reaction did not run to completion over the 1800 s of the experiment. Addition of higher quantities of iron resulted in the formation of a yellow/red precipitate at the end of the reaction. We also performed these assays on purified recombinant encapsulin; which, when assayed alone, did not display ferroxidase activity above background $Fe^{2+}$ oxidation (*Figure 8B*). In contrast, complexes of the full EncFtn encapsulin nanocompartment (i.e. the EncFtn-Enc protein complex) displayed ferroxidase activity comparable to apoferritin without the formation of precipitates (*Figure 8B*).

We attributed the precipitates observed in the EncFtn$_{sH}$ ferroxidase assay to the production of insoluble $Fe^{3+}$ complexes, which led us to propose that EncFtn does not directly store $Fe^{3+}$ in a mineral form. This observation agrees with native MS results, which indicates a maximum iron loading of 10–15 iron ions per decameric EncFtn; and the structure, which does not possess the enclosed iron-storage cavity characteristic of classical ferritins and Dps family proteins that can directly accrue mineralized $Fe^{3+}$ within their nanocompartment structures.

To analyze the products of these reactions and determine whether the EncFtn and encapsulin were able to store iron in a mineral form, we performed TEM on the reaction mixtures from the ferroxidase assay. The EncFtn$_{sH}$ reaction mixture showed the formation of large, irregular electron-dense precipitates (*Figure 8—figure supplement 1A*). A similar distribution of particles was observed after addition of $Fe^{2+}$ to the encapsulin protein (*Figure 8—figure supplement 1B*). In contrast, addition of $Fe^{2+}$ to the EncFtn-Enc nanocompartment resulted in small, highly regular, electron dense particles of approximately 5 nm in diameter (*Figure 8—figure supplement 1C*); we interpret these observations as controlled mineralization of iron within the nanocompartment. Addition of $Fe^{2+}$ to apoferritin resulted in a mixture of large particles and small (~2 nm) particles consistent with partial mineralization by the ferritin and some background oxidation of the iron (*Figure 8—figure supplement 1D*). Negative stain TEM of these samples revealed that upon addition of iron, the EncFtn$_{sH}$ protein showed significant aggregation (*Figure 8—figure supplement 1F*); while the encapsulin, EncFtn-Enc system, and apoferritin are present as distinct nanocompartments without significant protein aggregation (*Figure 8—figure supplement 1G–I*).

## Iron storage in encapsulin nanocompartments

The results of the ferroxidase assay and micrographs of the reaction products suggest that the oxidation and mineralization function of the classical ferritins are split between the EncFtn and encapsulin proteins, with the EncFtn acting as a ferroxidase and the encapsulin shell providing an environment and template for iron mineralization and storage. To investigate this further, we added $Fe^{2+}$ at various concentrations to samples of apo-ferritin, EncFtn, isolated encapsulin, and the EncFtn-Enc protein complex, and subjected these samples to a ferrozine assay to quantify the

amount of iron associated with the proteins after three hours of incubation. The maximum iron loading capacity of these systems was calculated as the quantity of iron per biological assembly (*Figure 8C*). In this assay, the EncFtn$_{sH}$ decamer binds a maximum of around 48 iron ions before excess iron induces protein precipitation. The encapsulin shell protein can sequester about 2200 iron ions before significant protein loss occurs, and the reconstituted EncFtn-Enc nanocompartment sequestered about 4150 iron ions. This latter result is significantly more than the apoferritin used in our assay, which sequesters approximately 570 iron ions in this assay (*Figure 8C*, *Table 5*).

Consideration of the functional oligomeric states of these proteins, where EncFtn is a decamer and encapsulin forms an icosahedral cage, and estimation of the iron loading capacity of these complexes gives insight into the role of the two proteins in iron storage and mineralization. EncFtn decamers bind up to 48 iron ions (*Figure 8C*), which is significantly higher than the stoichiometry of fifteen metal ions visible in the FOC and E31/34-site of the crystal structure of the EncFtn$_{sH}$ decamer and our MS analysis. The discrepancy between these solution measurements and our MS analysis may indicate that there are additional metal-binding sites on the interior channel and exterior faces of the protein; this is consistent with our identification of a number of weak metal-binding sites at the surface of the protein in the crystal structure (*Figure 5D*). These observations are consistent with hydrated $Fe^{2+}$ ions being channeled to the active site from the E31/34-site and the subsequent exit of $Fe^{3+}$ products on the outer surface, as is seen in other ferritin family proteins (*Pesek et al., 2011*; *Behera and Theil, 2014*). While the isolated encapsulin shell does not display any ferroxidase activity, it binds around 2200 iron ions in our assay (*Table 5*). This implies that the shell can bind a significant amount of iron on its outer and inner surfaces. While the maximum reported loading capacity of classical ferritins is approximately 4500 iron ions (*Mann et al., 1986*), in our assay system we were only able to load apoferritin with around 570 iron ions. However, the recombinant EncFtn-Enc nanocompartment was able to bind over 4100 iron ions in the same time period, over seven times the amount seen for the apoferritin. We note we do not reach the experimental maximum iron loading for apoferritin and therefore the total iron-loading capacity of our system may be significantly higher than in this experimental system.

Taken together, our data show that EncFtn can catalytically oxidize $Fe^{2+}$ to $Fe^{3+}$; however, iron binding in EncFtn is limited to the FOC and several surface metal binding sites. In contrast, the

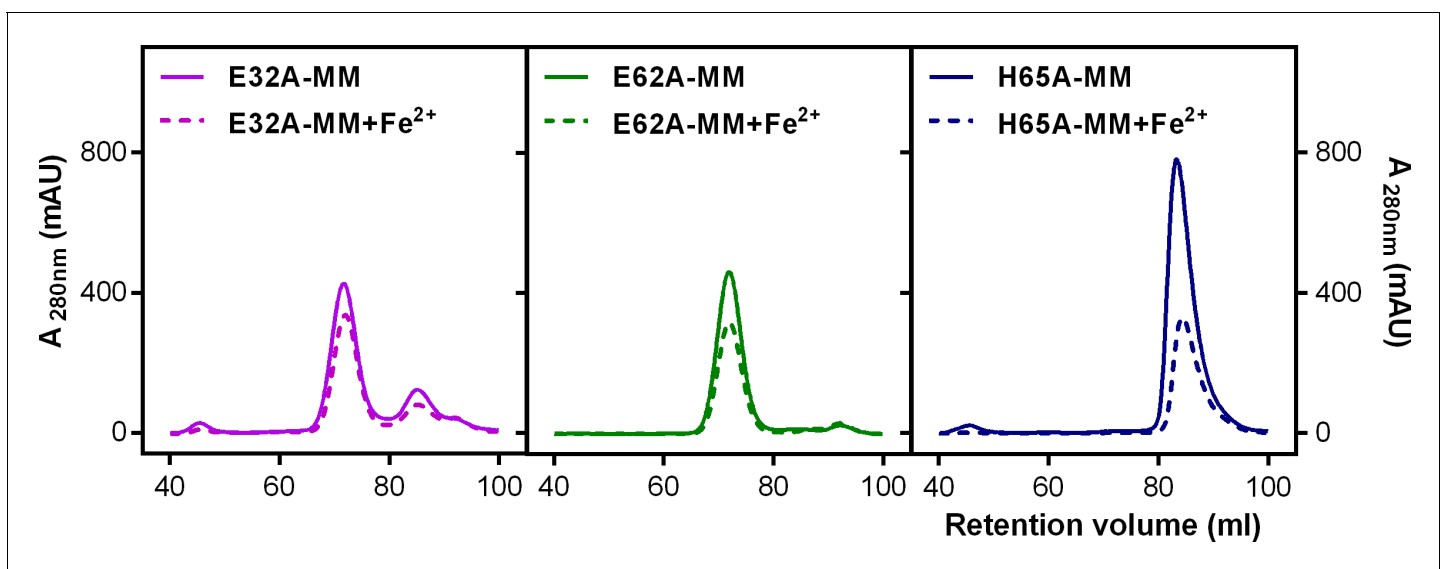

**Figure 9.** Purification of recombinant *R. rubrum* EncFtn$_{sH}$ FOC mutants. Single mutants E32A, E62A, and H65A of EncFtn$_{sH}$ produced from *E. coli* BL21 (DE3) cells grown in MM and MM supplemented with iron were subjected to Superdex 200 size-exclusion chromatography. (**A**) Gel-filtration chromatogram of the E32A mutant form of EncFtn$_{sH}$ resulted in an elution profile with a majority of the protein eluting as the decameric form of the protein and a small proportion of monomer. (**B**) Gel-filtration chromatograhy of the E62A mutant form of EncFtn$_{sH}$ resulted in an elution profile with a single major decameric peak. (**C**) Gel-filtration chromatography of the H65A mutant form of EncFtn$_{sH}$ resulted in a single peak corresponding to the protein monomer.

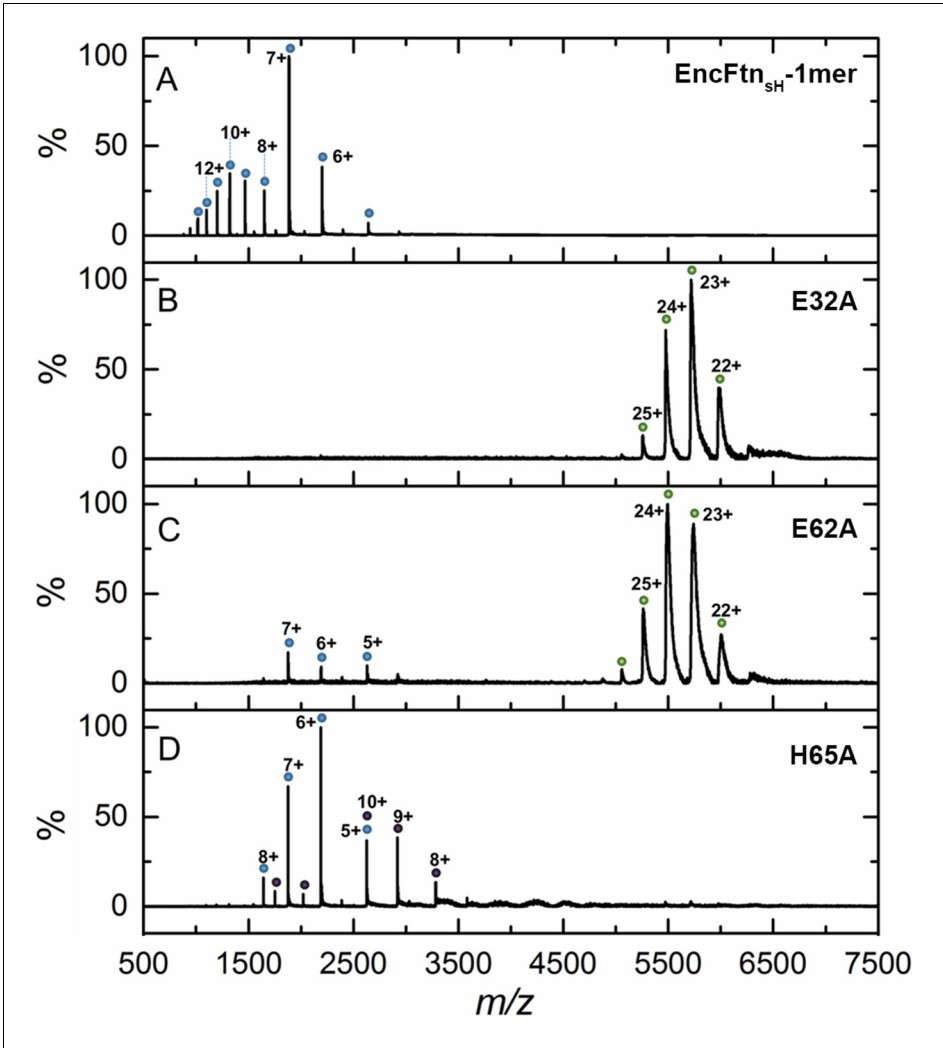

**Figure 10.** Native mass spectrometry of EncFtn_{sH} mutants. All spectra were acquired in 100 mM ammonium acetate, pH 8.0 with a protein concentration of 5 μM. (**A**) Wild-type EncFtn_{sH} in the absence of iron displays a charge state distribution consistent with a monomer (see also *Figure 8*). (**B**) E32A EncFtn_{sH} displays a charge states consistent with a decamer (green circles); a minor species, consistent with the monomer of E32A mutant is also observed (blue circles). (**C**) E62A EncFtn_{sH} displays charge states consistent with a decamer (green circles). (**D**) H65A EncFtn_{sH} displays charge states consistent with both monomer (blue circles) and dimer (purple circles).

encapsulin protein displays no catalytic activity, but has the ability to bind a considerable amount of iron. Finally, the EncFtn-Enc nanocompartment complex retains the catalytic activity of EncFtn, and sequesters iron within the encapsulin shell at a higher level than the isolated components of the system, and at a significantly higher level than the classical ferritins (*Andrews, 2010*). Furthermore, our recombinant nanocompartments may not have the physiological subunit stoichiometry, and the iron-loading capacity of native nanocompartments is potentially much higher than the level we have observed.

## Mutagenesis of the EncFtn_{sH} ferroxidase center

To investigate the structural and biochemical role played by the metal binding residues in the di-iron FOC of EncFtn_{sH} we produced alanine mutations in each of these residues: Glu32, Glu62, and His65. These EncFtn_{sH} mutants were produced in *E. coli* cells grown in MM, both in the absence and presence of additional iron. The E32A and E62A mutants eluted from SEC at a volume consistent with the decameric form of EncFtn_{sH}, with a small proportion of monomer; the H65A mutant eluted at a

**Table 4.** Data collection and refinement statistics. Statistics for the highest-resolution shell are shown in parentheses. Friedel mates were averaged when calculating reflection numbers and statistics.

| | WT | E32A | E62A | H65A |
|---|---|---|---|---|
| Data collection | | | | |
| Wavelength (Å) | 1.74 | 1.73 | 1.73 | 1.74 |
| Resolution range (Å) | 49.63 - 2.06 (2.10 - 2.06) | 48.84 - 2.59 (2.683 - 2.59) | 48.87 - 2.21 (2.29 - 2.21) | 48.86 - 2.97 (3.08 - 2.97) |
| Space group | $P\ 1\ 2_1\ 1$ | $P\ 1\ 2_1\ 1$ | $P\ 1\ 2_1\ 1$ | $P\ 1\ 2_1\ 1$ |
| Unit cell (Å) $a$ b c $\beta$ (°) | 98.18 120.53 140.30 95.36 | 97.78 120.28 140.53 95.41 | 98.09 120.23 140.36 95.50 | 98.03 120.29 140.43 95.39 |
| Total reflections | 1,264,922 (41,360) | 405,488 (36,186) | 1,069,345 (95,716) | 323,853 (32,120) |
| Unique reflections | 197,873 (8,766) | 100,067 (9,735) | 162,379 (15,817) | 66,658 (6,553) |
| Multiplicity | 6.4 (4.7) | 4.1 (3.7) | 6.6 (6.1) | 4.9 (4.9) |
| Anomalous multiplicity | 3.2 (2.6) | N/A | N/A | N/A |
| Completeness (%) | 99.2 (88.6) | 99.0 (97.0) | 100 (97.0) | 100 (99.0) |
| Anomalous completeness (%) | 96.7 (77.2) | N/A | N/A | N/A |
| Mean I/sigma(I) | 10.6 (1.60) | 8.46 (1.79) | 13.74 (1.80) | 8.09 (1.74) |
| Wilson B-factor | 26.98 | 40.10 | 33.97 | 52.20 |
| $R_{merge}$ | 0.123 (0.790) | 0.171 (0.792) | 0.0979 (1.009) | 0.177 (0.863) |
| $R_{meas}$ | 0.147 (0.973) | 0.196 (0.923) | 0.1064 (1.107) | 0.199 (0.966) |
| CC1/2 | 0.995 (0.469) | 0.985 (0.557) | 0.998 (0.642) | 0.989 (0.627) |
| CC* | 0.999 (0.846) | 0.996 (0.846) | 0.999 (0.884) | 0.997 (0.878) |
| Image DOI | 10.7488/ds/1342 | 10.7488/ds/1419 | 10.7488/ds/1420 | 10.7488/ds/1421 |
| Refinement | | | | |
| $R_{work}$ | 0.171 (0.318) | 0.183 (0.288) | 0.165 (0.299) | 0.186 (0.273) |
| $R_{free}$ | 0.206 (0.345) | 0.225 (0351) | 0.216 (0.364) | 0.237 (0.325) |
| Number of non-hydrogen atoms | 23,222 | 22,366 | 22,691 | 22,145 |
| macromolecules | 22,276 | 22,019 | 21,965 | 22,066 |
| ligands | 138 | 8 | 24 | 74 |
| water | 808 | 339 | 702 | 5 |
| Protein residues | 2,703 | 2,686 | 2,675 | 2,700 |
| RMS(bonds) (Å) | 0.012 | 0.005 | 0.011 | 0.002 |
| RMS(angles) (°) | 1.26 | 0.58 | 1.02 | 0.40 |
| Ramachandran favored (%) | 100 | 99 | 100 | 99 |
| Ramachandran allowed (%) | 0 | 1 | 0 | 1 |
| Ramachandran outliers (%) | 0 | 0 | 0 | 0 |
| Clash score | 1.42 | 1.42 | 1.79 | 0.97 |
| Average B-factor (Å$^2$) | 33.90 | 42.31 | 41.34 | 47.68 |
| macromolecules | 33.80 | 42.35 | 41.31 | 47.60 |
| ligands | 40.40 | 72.80 | 65.55 | 72.34 |
| solvent | 36.20 | 38.95 | 41.46 | 33.85 |
| PDB ID | 5DA5 | 5L89 | 5L8B | 5L8G |

**Table 5.** Iron loading capacity of EncFtn, encapsulin and ferritin. Protein samples (at 8.5 µM) including decameric EncFtn$_{sH}$, encapsulin, EncFtn-Enc and apoferritin were mixed with Fe(NH$_4$)$_2$(SO$_4$) (in 0.1% (v/v) HCl) of different concentrations in 50 mM Tris-HCl (pH 8.0), 150 mM NaCl buffer at room temperature for 3 hrs in the air. Protein-Fe mixtures were centrifuged at 13,000 x g to remove precipitated material and desalted prior to the Fe and protein content analysis by ferrozine assay and BCA microplate assay, respectively. Fe to protein ratio was calculated to indicate the Fe binding capacity of the protein. Protein stability was compromised at high iron concentrations; therefore, the highest iron loading with the least protein precipitation was used to derive the maximum iron loading capacity per biological assembly (underlined and highlighted in bold). The biological unit assemblies are a decamer for EncFtn$_{sH}$, a 60mer for encapsulin, a 60mer of encapsulin loaded with 12 copies of decameric EncFtn in the complex, and 24mer for horse spleen apoferritin. Errors are quoted as the standard deviation of three technical repeats in both the ferrozine and BCA microplate assays. The proteins used in Fe loading experiment came from a single preparation.

| Protein sample | Fe(NH$_4$)$_2$(SO$_4$)$_2$ loading (µM) | Fe detected by ferrozine assay (µM) | Protein detected by BCA microplate assay (µM) | Fe / monomeric protein | Maximum Fe loading per biological assembly unit |
|---|---|---|---|---|---|
| 8.46 µM EncFtn$_{sH}$-10mer | 0 | 4.73 ± 2.32 | 5.26 ± 0.64 | 0.90 ± 0.44 | |
| | 39.9 | 9.93 ± 1.20 | 5.36 ± 0.69 | 1.85 ± 0.22 | |
| | 84 | 17.99 ± 2.01 | 4.96 ± 0.04 | 3.63 ± 0.41 | |
| | 147 | 21.09 ± 1.94 | 4.44 ± 0.21 | 4.75 ± 0.44 | 48 ± 4 |
| | 224 | 28.68 ± 0.30 | 3.73 ± 0.53 | 7.68 ± 0.08 | |
| | 301 | 11.27 ± 1.10 | 2.50 ± 0.05 | 4.51 ± 0.44 | |
| 8.50 µM Encapsulin | 0 | -1.02 ± 0.54 | 8.63 ± 0.17 | -0.12 ± 0.06 | |
| | 224 | 62.24 ± 2.49 | 10.01 ± 0.58 | 6.22 ± 0.35 | |
| | 301 | 67.94 ± 3.15 | 8.69 ± 0.42 | 7.81 ± 0.36 | |
| | 450 | 107.96 ± 8.88 | 8.50 ± 0.69 | 12.71 ± 1.05 | |
| | 700 | 97.51 ± 3.19 | 7.26 ± 0.20 | 13.44 ± 0.44 | |
| | 1000 | 308.63 ± 2.06 | 8.42 ± 0.34 | 36.66 ± 0.24 | 2199 ± 15 |
| | 1500 | 57.09 ± 0.90 | 1.44 ± 0.21 | 39.77 ± 0.62 | |
| | 2000 | 9.2 ± 1.16 | 0.21 ± 0.14 | 44.73 ± 5.63 | |
| 8.70 µM EncFtn-Enc | 0 | 3.31 ± 1.57 | 6.85 ± 0.07 | 0.48 ± 0.23 | |
| | 224 | 116.27 ± 3.74 | 7.63 ± 0.12 | 15.25 ± 0.49 | |
| | 301 | 132.86 ± 4.03 | 6.66 ± 0.31 | 19.96 ± 0.61 | |
| | 450 | 220.57 ± 27.33 | 6.12 ± 1.07 | 36.06 ± 4.47 | |
| | 700 | 344.03 ± 40.38 | 6.94 ± 0.17 | 49.58 ± 5.82 | |
| | 1000 | 496.00 ± 38.48 | 7.19 ± 0.08 | 68.94 ± 5.35 | 4137 ± 321 |
| | 1500 | 569.98 ± 73.63 | 5.73 ± 0.03 | 99.44 ± 12.84 | |
| | 2000 | 584.30 ± 28.33 | 4.88 ± 0.22 | 119.62 ± 5.80 | |
| 8.50 µM Apoferritin | 0 | 3.95 ± 2.26 | 9.37 ± 0.24 | 0.42 ± 0.25 | |
| | 42.5 | 10.27 ± 1.12 | 8.27 ± 0.30 | 1.24 ± 0.18 | |
| | 212.5 | 44.48 ± 2.76 | 7.85 ± 0.77 | 5.67 ± 0.83 | |
| | 637.5 | 160.93 ± 4.27 | 6.76 ± 0.81 | 23.79 ± 3.12 | 571 ± 75 |
| | 1275 | 114.92 ± 3.17 | 3.84 ± 0.30 | 29.91 ± 2.95 | |
| | 1700 | 91.40 ± 3.37 | 3.14 ± 0.35 | 29.13 ± 3.86 | |

volume consistent with the monomeric form of EncFtn$_{sH}$ (*Figure 9*). For all of the mutants studied, no change in oligomerization state was apparent upon addition of Fe$^{2+}$ *in vitro*.

In addition to SEC studies, native mass spectrometry of the apo-EncFtn$_{sH}$ mutants was performed and compared with the wild-type apo-EncFtn$_{sH}$ protein (*Figure 10*). As described above, the apo-EncFtn$_{sH}$ has a charge state distribution consistent with an unstructured monomer, and decamer formation is only initiated upon addition of ferrous iron. Both the E32A mutant and E62A mutant displayed charge state distributions consistent with decamers, even in the absence of Fe$^{2+}$. This gas-

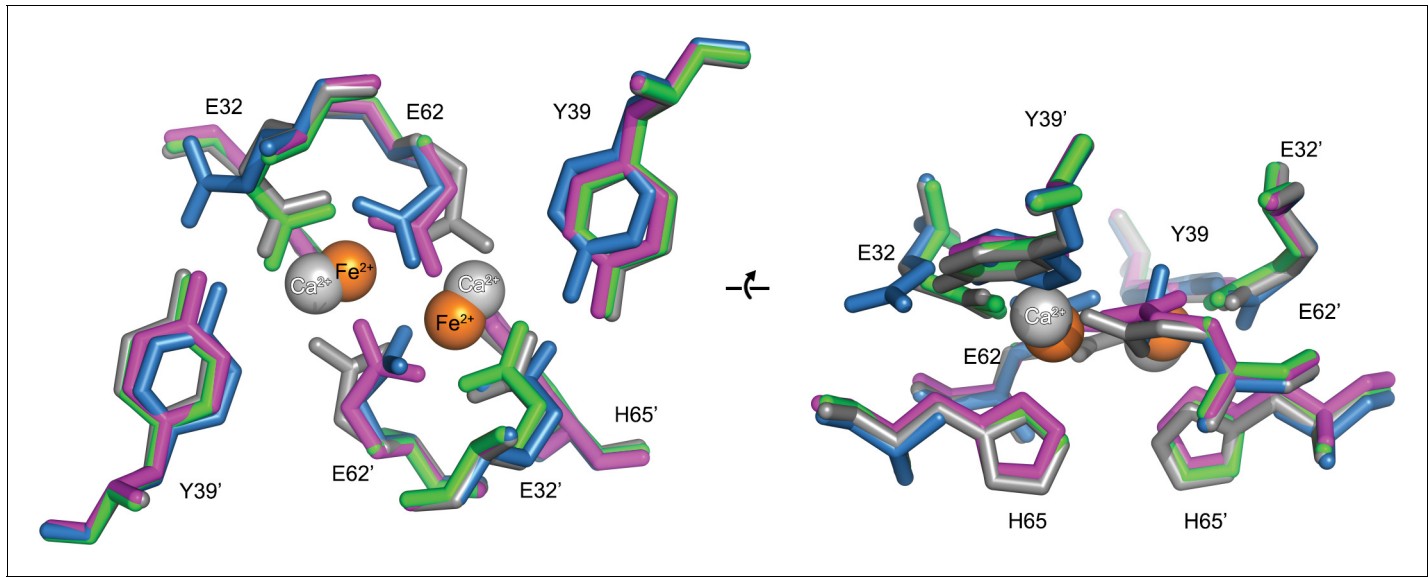

**Figure 11.** Comparison of the EncFtn$_{sH}$ FOC mutants *vs* wild type. The structures of the three EncFtn$_{sH}$ mutants were all determined by X-ray crystallography. The E32A, E62A and H65A mutants were crystallized in identical conditions to the wild type. EncFtn$_{sH}$ structure and were essentially isomorphous in terms of their unit cell dimensions. The FOC residues of the mutants and native EncFtn$_{sH}$ structures are shown as sticks with coordinated Fe$^{2+}$ as orange and Ca$^{2+}$ as grey spheres and are colored as follows: wild type, grey; E32A, pink; E62A, green; H65A, blue. Of the mutants, only H65A has any coordinated metal ions, which appear to be calcium ions from the crystallization condition. The overall organization of FOC residues is retained in the mutants, with almost no backbone movements. Significant differences center around Tyr39, which moves to coordinate the bound calcium ions in the H65A mutant; and Glu32, which moves away from the metal ions in this structure.

The following figure supplements are available for figure 11:

**Figure supplement 1.** FOC dimer interface of EncFtn$_{sH}$-E32A mutant.

**Figure supplement 2.** FOC dimer interface of EncFtn$_{sH}$-E62A mutant.

**Figure supplement 3.** FOC dimer interface of EncFtn$_{sH}$-H65A mutant.

phase observation is consistent with SEC measurements, which indicate both of these variants were also decamers in solution. Thus it seems that these mutations allow the decamer to form in the absence of iron in the FOC. In contrast to the glutamic acid mutants, MS analysis of the H65A mutant is similar to wild-type apo-EncFtn$_{sH}$ and is present as a monomer; interestingly a minor population of dimeric H65A was also observed.

We propose that the observed differences in the oligomerization state of the E32A and E62A mutants compared to wild-type are due to the changes in the electrostatic environment within the FOC. At neutral pH the glutamic acid residues are negatively charged, while the histidine residues are predominantly in their uncharged state. In the wild-type (WT) EncFtn$_{sH}$ this leads to electrostatic repulsion between subunits in the absence of iron. Coordination of Fe$^{2+}$ in this site stabilizes the dimer and reconstitutes the active FOC. The geometric arrangement of Glu32 and Glu62 in the FOC explains their behavior in solution and the gas phase, where they both favor the formation of decamers due to the loss of a repulsive negative charge. The FOC in the H65A mutant is destabilized through the loss of this metal coordinating residue and potential positive charge carrier, thus favoring the monomer in solution and the gas phase.

To understand the impact of the mutants on the organization and metal binding of the FOC, we determined the X-ray crystal structures of each of the EncFtn$_{sH}$ mutants (See *Table 4* for data collection and refinement statistics). The crystal packing of all of the mutants in this study is essentially isomorphous to the EncFtn$_{sH}$ structure. All of the mutants display the same decameric arrangement in the crystals as the EncFtn$_{sH}$ structure, and the monomers superimpose with an average RMSD$_{C\alpha}$ of less than 0.2 Å.

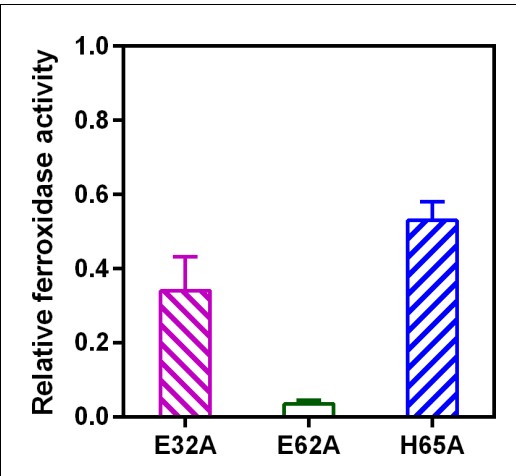

**Figure 12.** Relative ferroxidase activity of EncFtn$_{sH}$ mutants. EncFtn$_{sH}$, and the mutant forms E32A, E62A and H65A, each at 20 µM, were mixed with 100 µM acidic Fe(NH$_4$)$_2$(SO$_4$)$_2$. Ferroxidase activity of the mutant forms is determined by measuring the absorbance at 315 nm for 1800 s at 25 °C as an indication of Fe$^{3+}$ formation. The relative ferroxidase activity of mutants is plotted as a proportion of the activity of the wild-type protein using the endpoint measurement of A$_{315}$. Three technical repeats were performed and the plotted error bars represent the calculated standard deviations. The FOC mutants showed reduced ferroxidase activity to varied extents, among which E62A significantly abrogated the ferroxidase activity.

The following figure supplement is available for figure 12:

**Figure supplement 1.** Progress curves recording ferroxidase activity of EncFtn$_{sH}$ mutants. 20 µM wild-type EncFtn$_{sH}$, E32A, E62A and H65A mutants were mixed with 20 µM or 100 µM acidic Fe(NH$_4$)$_2$(SO$_4$)$_2$, respectively.

Close inspection of the region of the protein around the FOC in each of the mutants highlights their effect on metal binding (*Figure 11* and *Figure 11—figure supplement 1–3*). In the E32A mutant the position of the side chains of the remaining iron coordinating residues in the FOC is essentially unchanged, but the absence of the axial-metal coordinating ligand provided by the Glu32 side chain abrogates metal binding in this site. The Glu31/34-site also lacks metal, with the side chain of Glu31 rotated by 180° at the C$_\beta$ in the absence of metal (*Figure 11—figure supplement 1*). The E62A mutant has a similar effect on the FOC to the E32A mutant, however the entry site still has a calcium ion coordinated between residues Glu31 and Glu34 (*Figure 11—figure supplement 2*). The H65A mutant diverges significantly from the wild type in the position of the residues Glu32 and Tyr39 in the FOC. E32 appears in either the original orientation as the wild type and coordinates Ca$^{2+}$ in this position, or it is flipped by 180° at the C$_\beta$, moving away from the coordinated calcium ion in the FOC. Tyr39 moves closer to Ca$^{2+}$ compared to the wild-type and coordinates the calcium ion (*Figure 11—figure supplement 3*). A single calcium ion is present in the entry site of this mutant; however, Glu31 of one chain is rotated away from the metal ion and is not involved in coordination.

Taken together the results of our data show that these changes to the FOC of EncFtn still permit the formation of the decameric form of the protein. While the proteins all appear decameric in crystals, their solution and gas-phase behavior differs considerably and the mutants no longer show metal-dependent oligomerization. These results highlight the importance of metal coordination in the FOC for the stability and assembly of the EncFtn protein.

To address the question of how mutagenesis of the iron coordinating residues affects the enzymatic activity of the EncFtn$_{sH}$ protein we recorded progress curves for the oxidation of Fe$^{2+}$ to Fe$^{3+}$ by the different mutants as before. Mutagenesis of E32A and H65A reduces the activity of EncFtn$_{sH}$ by about 40%-55%; the E62A mutant completely abrogates activity, presumably through the loss of the bridging coordination for the formation of the di-nuclear iron center of the FOC (*Figure 12*). Collectively, the effect of mutating these residues in the FOC confirms the importance of the iron coordinating residues for the ferroxidase activity of the EncFtn$_{sH}$ protein.

## Discussion

Our study reports on a new class of ferritin-like proteins (EncFtn), which are associated with bacterial encapsulin nanocompartments (Enc). By studying the EncFtn from *R. rubrum* we demonstrate that iron binding results in assembly of EncFtn decamers, which display a unique annular architecture. Despite a radically different quaternary structure to the classical ferritins, the four-helical bundle scaffold and FOC of EncFtn$_{sH}$ are strikingly similar to ferritin (*Figure 6A*). A sequence-based

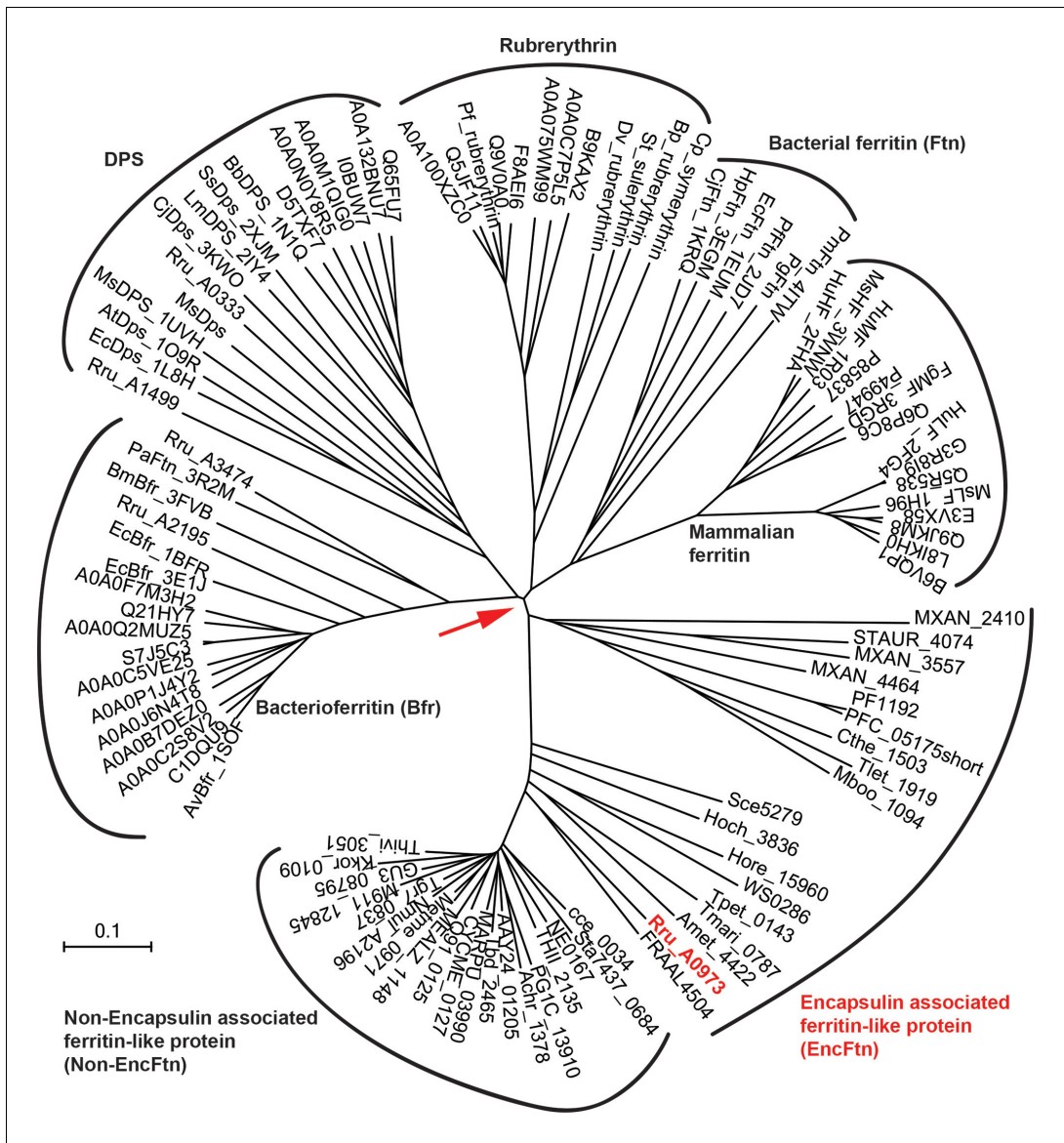

**Figure 13.** Phylogenetic tree of ferritin family proteins. The tree was built using the Neighbor-Joining method (*Saitou and Nei, 1987*) based on step-wise amino acid sequence alignment of the four-helical bundle portions of ferritin family proteins (*Supplementary file 1*). The tree is drawn to scale, with branch lengths in the same units as those of the evolutionary distances used to infer the phylogenetic tree; the likely root of the tree is indicated by a red arrow. The evolutionary distances were computed using the p-distance method (*Nei and Kumar, 2000*) and are in the units of the number of amino acid differences per site. The rate variation among sites was modeled with a gamma distribution (shape parameter = 2.5). The analysis involved 104 amino acid sequences. All ambiguous positions were removed for each sequence pair. There were a total of 262 positions in the final dataset. Evolutionary analyses were conducted in MEGA7 (*McCoy et al., 2007*)

phylogenetic tree for proteins in the ferritin family was constructed; in addition to the classical ferritins, bacterioferritins and Dps proteins, our analysis included the encapsulin-associated ferritin-like proteins (EncFtns) and a group related to these, but lacking the encapsulin sequence (Non-EncFtn). The analysis revealed that the EncFtn and Non-EncFtn proteins form groups distinct from the other clearly delineated groups of ferritins, and represent outliers in the tree (*Figure 13*). While it is difficult to infer ancestral lineages in protein families, the similarity seen in the active site scaffold of these proteins highlights a shared evolutionary relationship between EncFtn proteins and other

members of the ferritin superfamily that has been noted in previous studies (*Andrews, 2010*; *Lundin et al., 2012*). From this analysis, we propose that the four-helical fold of the classical ferritins may have arisen through gene duplication of an ancestor of EncFtn. This gene duplication would result in the C-terminal region of one EncFtn monomer being linked to the N-terminus of another and thus stabilizing the four-helix bundle fold within a single polypeptide chain (*Figure 6B*). Linking the protein together in this way relaxes the requirement for the maintenance of a symmetrical FOC and thus provides a path to the diversity in active-site residues seen across the ferritin family (*Figure 6A*, residues Glu95, Gln128 and Glu131 in PmFtn, *Supplementary file 1*) (*Andrews, 2010*; *Lundin et al., 2012*).

## Relationship between ferritin structure and activity

The quaternary arrangement of classical ferritins into an octahedral nanocage and Dps into a dodecamer is absolutely required for their function as iron storage compartments (*Chasteen and Harrison, 1999*). The oxidation and mineralization of iron must be spatially separated from the host cytosol to prevent the formation of damaging hydroxyl radicals in the Fenton and Haber-Weiss reactions (*Honarmand Ebrahimi et al., 2012*). This is achieved in all ferritins by confining the oxidation of iron to the interior of the protein complex, thus achieving sequestration of the $Fe^{3+}$ mineralization product. A structural alignment of the FOC of EncFtn with the classical ferritin PmFtn shows that the central ring of EncFtn corresponds to the external surface of ferritin, while the outer circumference of EncFtn is congruent with the inner mineralization surface of ferritin (*Figure 6—figure supplement 1A*). This overlay highlights the fact that the ferroxidase center of EncFtn faces in the opposite direction relative to the classical ferritins and is essentially inside out regarding iron storage space (*Figure 6—figure supplement 1B*, boxed region). Analysis of each of the single mutations (E32A, E62A and H65A) made in the FOC highlights the importance of the iron-coordinating residues in the catalytic activity of EncFtn. Furthermore, the position of the calcium ion coordinated by Glu31 and Glu34 seen in the EncFtn$_{sH}$ structure suggests an entry site to channel metal ions into the FOC; we propose that this site binds hydrated iron ions *in vivo* and acts as a selectivity filter and gate for the FOC (*Haldar et al., 2011*). The constellation of charged residues on the outer circumference of EncFtn (His57, Glu61 and Glu64) could function in the same way as the residues lining the mineralization surface within the classical ferritin nanocage (*Le Brun et al., 2010*), and given their proximity to the FOC these sites may be the exit portal and mineralization site (*Honarmand Ebrahimi et al., 2012*).

The absolute requirement for the spatial separation of oxidation and mineralization in ferritins suggests that the EncFtn family proteins are not capable of storing iron minerals due to the absence of an enclosed compartment in their structure (*Figure 6—figure supplement 1B*). Our biochemical characterization of EncFtn supports this hypothesis, indicating that while this protein is capable of oxidizing iron, it does not accrue mineralized iron in an analogous manner to classical ferritins. While EncFtn does not store iron itself, its association with the encapsulin nanocage suggests that mineralization occurs within the cavity of the encapsulin shell (*McHugh et al., 2014*). Our ferroxidase assay data on the recombinant EncFtn-Enc nanocompartments, which accrue over 4100 iron ions per complex and form regular nanoparticles, are consistent with the encapsulin protein acting as the store for iron oxidized by the EncFtn enzyme. TEM analysis of the reaction products shows the production of homogeneous iron nanoparticles only in the EncFtn-Enc nanocompartment (*Figure 8—figure supplement 1*).

Docking the decamer structure of EncFtn$_{sH}$ into the pentamer of the *T. maritima* encapsulin Tmari_0786 (PDB ID: 3DKT) (*Sutter et al., 2008*) shows that the position of the C-terminal extensions of our EncFtn$_{sH}$ structure are consistent with the localization sequences seen bound to the encapsulin protein (*Figure 14A*). Thus, it appears that the EncFtn decamer is the physiological state of this protein. This arrangement positions the central ring of EncFtn directly above the pore at the five-fold symmetry axis of the encapsulin shell and highlights a potential route for the entry of iron into the encapsulin and towards the active site of EncFtn. A comparison of the encapsulin nanocompartment and the ferritin nanocage highlights the size differential between the two complexes (*Figure 14B*) that allows the encapsulin to store significantly more iron. The presence of five FOCs per EncFtn$_{sH}$ decamer and the fact that the icosahedral encapsulin nanocage can hold up to twelve of decameric EncFtn between each of the internal five-fold vertices means that they can achieve a high rate of iron mineralization across the entire nanocompartment. This arrangement of multiple

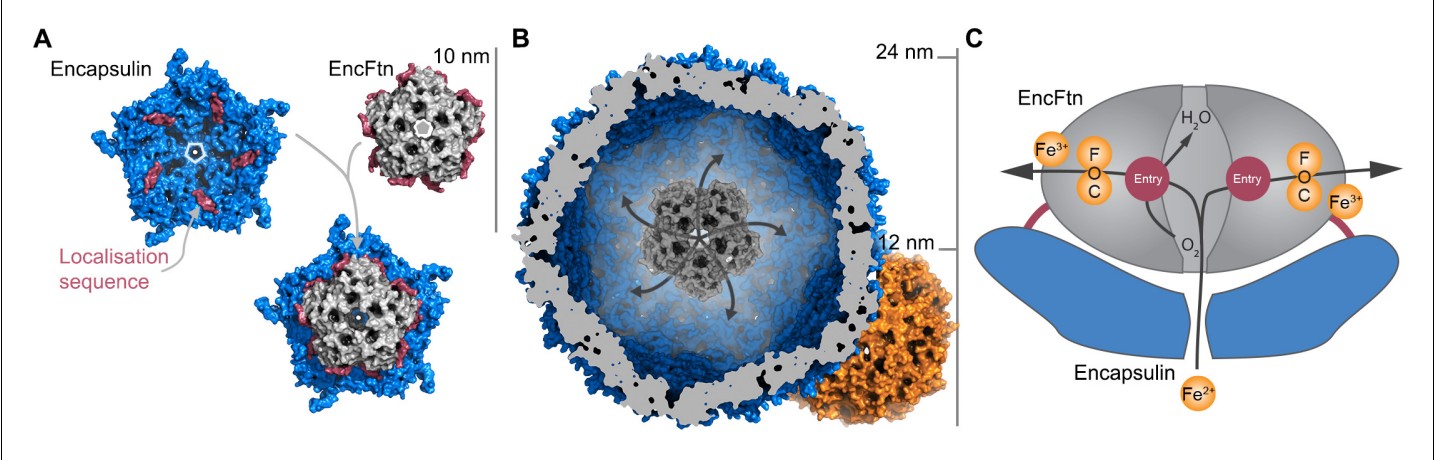

**Figure 14.** Model of iron oxidation in encapsulin nanocompartments. (**A**) Model of EncFtnsH docking to the encapsulin shell. A single pentamer of the icosahedral *T. maritima* encapsulin structure (PDBID: 3DKT) (*Sutter et al., 2008*) is shown as a blue surface with the encapsulin localization sequence of EncFtn shown as a purple surface. The C-terminal regions of the EncFtn subunits correspond to the position of the localization sequences seen in 3DKT. Alignment of EncFtnsH with 3DKT positions the central channel directly above the pore in the 3DKT pentamer axis (shown as a grey pentagon). (**B**) Surface view of EncFtn within the encapsulin nanocompartment (grey and blue respectively). The lumen of the encapsulin nanocompartment is considerably larger than the interior of ferritin (shown in orange behind the encapsulin for reference) and thus allows the storage of significantly more iron. The proposed pathway for iron movement through the encapsulin shell and EncFtn FOC is shown with arrows. (**C**) Model ofiron oxidation within an encapsulin nanocompartment. As EncFtn is unable to mineralize iron on its surface directly, $Fe^{2+}$ must pass through the encapsulin shell to access the first metal binding site within the central channel of EncFtnsH (entry site) prior to oxidation within the FOC and release as $Fe^{3+}$ to the outer surface of the protein where it can be mineralized within the lumen of the encapsulin cage.

reaction centers in a single protein assembly is reminiscent of classical ferritins, which has 24 FOCs distributed around the nanocage.

Our structural data, coupled with biochemical and ICP-MS analysis, suggest a model for the activity of the encapsulin iron-megastore (*Figure 14C*). The crystal structure of the *T. maritima* encapsulin shell protein has a negatively charged pore positioned to allow the passage of $Fe^{2+}$ into the encapsulin and directs the metal towards the central, negatively charged hole of the EncFtn ring (*Figure 4—figure supplement 1*). The five metal-binding sites on the interior of the ring (Glu31/34-sites) may select for the $Fe^{2+}$ ion and direct it towards their cognate FOCs. We propose that the oxidation of $Fe^{2+}$ to $Fe^{3+}$ occurs within the FOC according to the model postulated by (*Honarmand Ebrahimi et al., 2012*) in which the FOC acts as a substrate site through which iron passes and is released on to weakly coordinating sites at the outer circumference of the protein (His57, Glu61 and Glu64), where it is able to form ferrihydrite minerals which can be safely deposited within the lumen of the encapsulin nanocompartment (*Figure 14*).

Here we describe for the first time the structure and biochemistry of a new class of encapsulin-associated ferritin-like protein and demonstrate that it has an absolute requirement for compartmentalization within an encapsulin nanocage to act as an iron store. Further work on the EncFtn-Enc nanocompartment will establish the structural basis for the movement of iron through the encapsulin shell, the mechanism of iron oxidation by the EncFtn FOC and its subsequent storage in the lumen of the encapsulin nanocompartment.

# Materials and methods

## Cloning

Genes of interest were amplified by PCR using *R. rubrum* ATCC 11,170 genomic DNA (DSMZ) as the template and KOD Hot Start DNA Polymerase (Novagen). Primers used in this study are listed in *Supplementary file 2*. PCR products were visualized in 0.8% agarose gel stained with SYBR Safe (Life Technologies, UK). Fragments of interest were purified by gel extraction (Qiagen, UK) before digestion by endonuclease restriction enzymes (Thermo Fisher Scientific, UK) at 37°C for 1 hr,

followed by ligation with similarly digested vector pET-28a(+) or pACYCDuet-1 at room temperature for 1 hr. Ligation product was transformed into chemically competent *Escherichia coli* Top10 cells and screened against 50 ng/μl kanamycin for pET-28a(+) based constructs or 34 ng/μl chloramphenicol for pACYCDuet-1 based constructs. DNA insertion was confirmed through Sanger sequencing (Edinburgh Genomics, The University of Edinburgh, UK). Sequence verified constructs were transformed into *E. coli* BL21(DE3) or Tuner(DE3) for protein production. Alternatively, plasmids transformed into *E. coli* B834(DE3) cells were cultured in selenomethionine medium.

## Protein production and purification

A single colony of *E. coli* BL21(DE3) or Tuner(DE3) cells, transformed with protein expression plasmid, was transferred into 10 ml LB medium, or M9 minimal medium (MM), supplemented with appropriate antibiotic, and incubated overnight at 37 °C with 200 rpm shaking. The overnight pre-culture was then inoculated into 1 liter of LB medium and incubated at 37 °C with 200 rpm shaking. Recombinant protein production was induced at $OD_{600}= 0.6$ by the addition of 1 mM IPTG and the incubation temperature was reduced to 18°C for overnight incubation. Cells were pelleted by centrifugation at 4000 *g* for 20 min at 4 °C, and resuspended 10-fold (volume per gram of cell pellet) in PBS to wash cells before a second centrifugation step. Cells were resuspended in 10-times (v/w) of appropriate lysis buffer for the purification method used (see details of buffers below) and lysed by sonication on ice, with ten cycles of 30-second burst of sonication at 10 μm amplitude and 30 s of cooling. Cell lysate was clarified by centrifugation at 20,000 *x g*, 30 min, 4 °C; followed by filtration using a 0.22 μM syringe filter (Millipore, UK).

Selenomethionine labelled protein was produced by growing a single colony of *E. coli* B834 (DE3) cells transformed with protein expression plasmids in 100 ml LB medium supplemented with appropriate antibiotic overnight at 37 °C with shaking at 200 rpm. The overnight pre-culture was pelleted by centrifugation 3,000 *x g*, 4 °C, 15 min and washed twice with M9 minimal medium. The washed cells were transferred to 1 liter of SeMet medium, which contains M9 minimal medium, 40 mg/L of each ʟ-amino acid (without methionine), 40 mg/L selenomethionine, 2 mM $MgSO_4$, 0.4% (w/v) glucose and 1 mM $Fe(NH_4)_2(SO_4)_2$. Cells were incubated at 37 °C with 200 rpm shaking and recombinant protein production was induced at $OD_{600}= 0.6$ by the addition of 1 mM IPTG and the incubation temperature was reduced to 18 °C for overnight incubation. Cells were harvested and lysed as above.

## His-tagged protein purification

Clarified cell lysate was loaded onto a 5 ml HisTrap column (GE Healthcare, UK) pre-equilibrated with HisA buffer (50 mM Tris-HCl, 500 mM NaCl and 50 mM imidazole, pH 8.0). Unbound proteins were washed from the column with HisA buffer. His-tagged proteins were then eluted by a step gradient of 50% HisA buffer and 50% HisB buffer (50 mM Tris-HCl, 500 mM NaCl and 500 mM imidazole, pH 8.0). Fractions containing the protein of interest, as determined by 15% (w/v) acrylamide SDS-PAGE, were pooled before loading onto a gel-filtration column (HiLoad 16/600 Superdex 200, GE Healthcare) equilibrated with GF buffer (50 mM Tris-HCl, pH 8.0, 150 mM NaCl). Fractions were subjected to 15% SDS-PAGE and those containing the protein of interest were pooled for further analysis.

## Sucrose gradient ultracentrifugation purification

Co-expressed encapsulin and EncFtn (EncFtn-Enc) and encapsulin protein were both purified according to the protocol used by M. Sutter (*Sutter, 2008*). Briefly, EncFtn-Enc or encapsulin was expressed based on pACYCDuet-1 vector. The *E. coli* cells were grown, induced, harvested and sonicated in a similar way as described above. GF buffer used in this purification contains 50 mM Tris-HCl, pH 8.0, and 150 mM NaCl. To remove RNA contamination, the lysate was supplemented with 50 μg/ml RNase A and rotated at 10 rpm and room temperature for 2 hrs, followed by centrifugation at 34,000 *x g* and 4 °C for 20 min and filtering through 0.22 μM syringe filter. Proteins were pelleted through 38% (w/v) sucrose cushion by ultracentrifugation at 100,000 *x g* and 4 °C for 21 hrs. 10% - 50% (w/v) sucrose gradient ultracentrifugation was applied to further separate the proteins at 100,000 *x g* and 4 °C for 17 hrs. Protein was dialyzed against GF buffer to remove sucrose before being used in chemical assays or TEM.

## Transmission electron microscopy

TEM imaging was performed on purified encapsulin, EncFtn, and EncFtn-Enc and apoferritin. Purified protein at 0.1 mg/ml concentration was spotted on glow-discharged 300 mesh carbon-coated copper grids and excess liquid wicked off with filter paper (Whatman, UK). The grids were washed with distilled water and blotted with filter paper three times before staining with 0.2% uranyl acetate, blotting and air-drying. Grids were imaged using a JEM1400 transmission electron microscope and images were collected with a Gatan CCD camera. Images were analyzed using ImageJ (NIH, Bethesda, MD) and size-distribution histograms were plotted using Prism 6 (GraphPad software). To observe iron mineral formation by TEM, protein samples at 8.5 µM concentration including EncFtn$_{sH}$, encapsulin, EncFtn-Enc and apoferritin were supplemented with acidic Fe(NH$_4$)$_2$(SO$_4$)$_2$ at their maximum iron loading ratio in room temperature for 1 hr. The mixtures were subjected to TEM analysis with or without uranyl acetate staining. TEM experiments without Fe loading were repeated three times, a representative set of images are presented here. Proteins loaded with Fe and imaged by TEM were from single preparation.

## Protein crystallization and X-ray data collection

EncFtn$_{sH}$ was purified by anion exchange and Superdex 200 size- exclusion chromatography and concentrated to 10 mg/ml (based on extinction coefficient calculation). Crystallization drops were set up using the hanging drop vapor diffusion method at 292 K. Glass coverslips were set up with 1–2 µl protein mixed with 1 µl well solution (0.14 M calcium acetate and 15% (w/v) PEG 3350) and sealed over 1 ml of well solution. Crystals appeared after 5 days and were harvested from the well using a LithoLoop (Molecular Dimensions Limited, UK), transferred briefly to a cryoprotection solution containing well solution supplemented with 1 mM FeSO$_4$ (in 0.1% (v/v) HCl), 20% (v/v) PEG 200, and subsequently flash cooled in liquid nitrogen. Crystals of the EncFtn$_{sH}$ single mutations were produced in the same manner as for the EncFtn$_{sH}$ wild-type protein.

All crystallographic datasets were collected on the macromolecular crystallography beamlines at Diamond Light Source (Didcot, UK) at 100 K using Pilatus 6M detectors. Diffraction data were integrated and scaled using XDS (*Kabsch, 2010*) and symmetry related reflections were merged with Aimless (*Evans, 2011*). Data collection statistics are shown in *Table 4*. The resolution cut-off used for structure determination and refinement was determined based on the CC$_{1/2}$ criterion proposed by *Karplus and Diederichs (2012)*.

The structure of EncFtn$_{sH}$ was determined by molecular replacement using PDB ID: 3K6C as the search model, modified to match the sequence of the target protein using Chainsaw (*Stein, 2008*). A single solution comprising three decamers in the asymmetric unit was found by molecular replacement using Phaser (*McCoy et al., 2007*). The initial model was rebuilt using Phenix.autobuild (*Adams et al., 2010*) followed by cycles of refinement with Phenix.refine (*Afonine et al., 2012*), with manual rebuilding and model inspection in Coot (*Emsley et al., 2010*). The final model was refined with isotropic B-factors, torsional NCS restraints, and with anomalous group refinement. The model was validated using MolProbity (*Chen et al., 2010*). Structural superimpositions were calculated using Coot. Crystallographic figures were generated with PyMOL. Multiple sequence alignment of EncFtn and ferritin family proteins was performed using Clustal Omega Sievers and Higgins, 2014 and displayed with Espript 3.0 (*Gouet et al., 2003*). Model refinement statistics are shown in *Table 4*. The final models and experimental data are deposited in the PDB and diffraction image files are available at the Edinburgh DataShare repository.

## Horse spleen apoferritin preparation

Horse spleen apoferritin purchased from Sigma Aldrich (UK) was dissolved in deaerated MOPS buffer (100 mM MOPS, 100 mM NaCl, 3 g/100 ml Na$_2$S$_2$O$_4$ and 0.5 M EDTA, pH 6.5) (*Bauminger et al., 1991*). Protein was dialyzed against 1 liter MOPS buffer in room temperature for two days before buffer exchanging to GF buffer (50 mM Tris-HCl, pH 8.0, 150 mM NaCl) in a vivaspin column with 5 kDa cut-off (Sartorius, UK) for several times. Fe content of apoferritin was detected using ferrozine assay (*Riemer et al., 2004*). Protein concentration was determined using Pierce Microplate BCA Protein Assay Kit. Apoferritin containing less than 0.5 Fe per 24-mer was used in the ferroxidase assay. Apoferritin used in the Fe loading capacity experiment was prepared in the same way with 5–15 Fe per 24-mer.

## Ferroxidase assay

1 mM and 200 µM $Fe(NH_4)_2(SO_4)_2$ stock solutions were prepared in 0.1% (v/v) HCl anaerobically. Protein solutions with 20 µM FOC were diluted from ~10 mg/ml frozen stock in GF buffer (50 mM Tris-HCl, pH 8.0 and 150 mM NaCl) anaerobically. Ferroxidase activity was initiated by adding 450 µl protein to 50 µl of acidic $Fe(NH_4)_2(SO_4)_2$ at the final concentration of 100 µM and 20 µM in the air, respectively. The ferroxidase activity was measured by monitoring the $Fe^{3+}$ formation which gives rise to the change of the absorbance at 315 nm (*Bonomi et al., 1996*). Absorbance at 315 nm was recorded every second over 1800 s using a quartz cuvette in a JASCO V-730 UV/VIS spectrophotometer (JASCO Inc., Easton, MD). In recombinantly coexpressed nanocompartments the ratio of EncFtn to Enc was assumed as 2 to 1, assuming each of the twelve pentameric vertices of the icosahedral encapsulin were occupied with decameric EncFtn. The data are presented as the mean of three technical replicates with error bars indicating one standard deviation from the mean. Proteins used here were from a single preparation.

## Iron loading capacity of ferritins

In order to determine the maximum iron loading capacity, around 8.5 µM proteins including decameric EncFtn$_{sH}$, Encapsulin, EncFtn-Enc and apoferritin were loaded with various amount of acidic $Fe(NH_4)_2(SO_4)_2$ ranging from 0 to 1700 µM. Protein mixtures were incubated in room temperature for 3 hrs before desalting in Zebra spin desalting columns (7 kDa cut-off, Thermo Fisher Scientific, UK) to remove free iron ions. The protein concentration was determined using PierceMicroplate BCA assay kit (Thermo Fisher Scientific). The protein standard curve was plotted according to the manufacturer. The Fe content in the samples was determined using modified ferrozine assay (*Riemer et al., 2004*). Briefly speaking, 100 µl protein sample was mixed with 100 µl mixture of equal volume of 1.4 M HCl and 4.5% (w/v) $KMnO_4$ and incubated at 60 °C for 2 hrs. 20 µl of the iron-detection reagent (6.5 mM ferrozine, 6.5 mM neocuproine, 2.5 M ammonium acetate, and 1 M ascorbic acid dissolved in $H_2O$) was added to the cooled tubes. 30 min later, 200 µl of the solution was transferred into a well of 96-well plate and the absorbance at 562 nm was measured on the plate reader Spectramax M5 (Molecular Devices, UK). The standard curve was plotted using various concentrations of $FeCl_3$ (in 10 mM HCl) diluted in the gel-filtration buffer. Three technical repeats were performed for both the ferrozine and microplate BCA assays. Samples analyzed by ICP-MS were prepared in the same way by mixing protein and ferrous ions and desalting. The proteins used in the Fe loading experiment came from a single preparation.

## Peroxidase assay

The peroxidase activity of EncFtn$_{sH}$ was determined by measuring the oxidation of *ortho*-phenylenediamine (OP) by $H_2O_2$ *Pesek et al. (2011)*. EncFtn$_{sH}$ decameric and monomeric fractions purified from MM were both used in the assay. *Ortho*-phenylenediamine was prepared as a 92.5 mM stock solution in 50 mM Tris-HCl (pH 8.0). 80, 70, 60, 50, 40, 30, 20 and 10 mM of OP were prepared by diluting the stock solution in the 50 mM Tris-HCl (pH 8.0). 100 µl of each diluted OP was added to a 96-well plate in 3 repeats. 1 µl of 32 µM protein was supplemented into each well to a final concentration of 160 nM, followed by the addition of 2 µl of 30% $H_2O_2$. After 15 min shaking in the dark, the reaction was stopped by adding 100 µl of 0.5 M $H_2SO_4$. The peroxidase activity was measured by monitoring the absorbance at 490 nm in the SpectraMax M5 Microplate Reader (Molecular Devices) (*Pesek et al., 2011*).

## ICP-MS analysis

Protein samples were diluted 50-fold into a solution of 2.5% $HNO_3$ (Suprapur, Merck, UK) containing 20 µg/L Pt as internal standard. Matrix-matched elemental standards (containing analyte metal concentrations 0 – 1000 µg/L) were prepared by serial dilution from individual metal standard stocks (VWR) with identical solution compositions, including the internal standard. All standards and samples were analyzed by ICP-MS using a Thermo x-series instrument (Thermo Fisher Scientific) operating in collision cell mode (using 3.0 ml min$^{-1}$ flow of 8% $H_2$ in He as the collision gas). Isotopes $^{44}$Ca, $^{56}$Fe, $^{66}$Zn, $^{78}$Se, and $^{195}$Pt were monitored using the peak-jump method (100 sweeps, 25–30 ms dwell time on 5 channels per isotope, separated by 0.02 atomic mass units) in triplicate. The protein samples used in ICP-MS came from a single protein preparation.

## Mass spectrometry analysis

For native MS analysis, all protein samples were buffer exchanged into 100 mM ammonium acetate (pH 8.0; adjusted with dropwise addition of 1% ammonia solution) using Micro Biospin Chromatography Columns (Bio-Rad, UK) prior to analysis and the resulting protein samples were analyzed at a final concentration of ~5 μM (oligomer concentration). In order to obtain Fe-bound EncFtn, 100 μM or 300 μM of freshly prepared $FeCl_2$ was added to apo-EncFtn$_{sH}$ (monomer peak) immediately prior to buffer exchange into 100 mM ammonium acetate (pH 8.0). Samples were analyzed on a quadrupole ion-mobility time of flight instrument (Synapt G2, Waters Corp., Manchester, UK), equipped with a nanomate nanoelectrospray infusion robot (Advion Biosciences, Ithaca, NY). Instrument parameters were tuned to preserve non-covalent protein complexes. After optimization, typical parameters were: nanoelectrospray voltage 1.54 kV; sample cone 50 V; extractor cone 0 V; trap collision voltage 4 V; source temperature 80°C; and source backing pressure 5.5 mbar. For improved mass resolution the sample cone was raised to 155 V. Ion mobility mass spectrometry (IM-MS) was performed using the travelling-wave mobility cell in the Synapt G2, employing nitrogen as the drift gas. Typically, the IMS wave velocity was set to 300 m/s; wave height to 15 V; and the IMS pressure was 1.8 mbar. All native MS experiments were performed on samples from two independent protein preparations. For collision cross section determination, IM-MS data was calibrated using denatured equine myoglobin and data was analyzed using Driftscope v2.5 and MassLynx v4.1 (Waters Corp., UK). Theoretical collision cross sections (CCS) were calculated from pdb files using IMPACT software v. 0.9.1 (*Marklund, 2015*). In order to obtain information on the topology of the EncFtn$_{sH}$ assembly, gas-phase dissociation of the Fe-associated EncFtn$_{sH}$ complex was achieved by increasing the sample cone and/or trap collision voltage prior to MS analysis.

## SEC-MALLS

Size-exclusion chromatography (ÄKTA-Micro; GE Healthcare) coupled to UV, static light scattering and refractive index detection (Viscotec SEC-MALS 20 and Viscotec RI Detector:VE3580; Malvern Instruments, UK) were used to determine the molecular mass of fractions decamer and monomer of EncFtn$_{sH}$ in solution individually. Protein concentration was determined by measurement of absorbance at 280 nm and calculated using the extinction coefficient ε0.1%= 1.462 mg$^{-1}$ ml$^{-1}$ cm$^{-1}$. 100 μl of 1.43 mgml$^{-1}$ fractions of EncFtn$_{sH}$ decamer and 4.03 mg ml$^{-1}$ fractions of EncFtn$_{sH}$ monomer were run individually on a Superdex 200 10/300 GL size-exclusion column pre-equilibrated in 50 mM Tris-HCl (pH 8.0), 150 mM NaCl at 22°C with a flow rate of 0.5 ml/min. Light scattering, refractive index (RI) and A$_{280nm}$ were analyzed by a homo-polymer model (OmniSEC software, v 5.1; Malvern Instruments) using the following parameters for fractions of decamer and monomer: the extinction coefficient (dA/dc) at 280 nm was 1.46 AU mg ml$^{-1}$ and specific refractive index increment (dn/dc) was 0.185 ml g$^{-1}$. The proteins analyzed by SEC-MALLS came from single protein preparation.

## Metal binding analysis by PAGE

Recombinant EncFtn$_{sH}$ fractions at 50 μM concentration were incubated with one molar equivalent of metal ions at room temperature for 2 hrs. Half of each sample was mixed with 5 x native loading buffer (65 mM Tris-HCl, pH 8.5, 20% glycerol and 0.01% bromophenol blue) and run on non-denaturing PAGE gels (10% acrylamide) and run in Tris/glycine buffer, 200 V, 4 °C for 50 min. The remaining samples were left for an additional three hours prior to SDS-PAGE (15% acrylamide) analysis. SDS-PAGE gels were run at room temperature at 200 V, room temperature for 50 min. Gels were stained with Coomassie Brilliant Blue R250 and scanned after de-staining in water. The proteins used in this experiment came from single protein preparation.

## Analytical size-exclusion chromatography

For analysis of the multimeric state of EncFtn proteins by analytical size-exclusion gel-filtration chromatography (AGF) 25 μl of 90 μM protein was loaded into Superdex 200 PC 3.2/30 column (GE Healthcare) at 15 °C with GF buffer running at 0.05 ml/min and pressure limit 0.45 MPa. In order to use AGF to determine how metal ions influence the assembly of EncFtn$_{sH}$, 90 μM EncFtn$_{sH}$ monomer fractions were mixed with equal molar concentrations of metal ion solutions including $FeSO_4$ in 0.1% (v/v) HCl, Fe(NH$_4$)$_2$(SO$_4$)$_2$, $FeCl_3$, $CoCl_2$, calcium acetate (CaAc), $ZnSO_4$ and $MnCl_2$ at room temperature for 2 hrs prior to AGF analysis. Protein samples without metal titration were also

analyzed as a control group. Both monomer and decamer fractions of EncFtn$_{sH}$ left at room temperature for 2 hrs, or overnight, were also analysed as controls to show the stability of the protein samples in the absence of additional metal ions. The AGF results have been repeated twice using two independent preparations of protein, of which only one representative trace is presented in the paper.

### Accession codes and datasets

Coordinates and structure factors for the structures presented in this paper have been deposited in the PDB under the following accession codes: EncFtn$_{sH}$, 5DA5; EncFtn$_{sH}$-E32A, 5L89; EncFtn$_{sH}$-E62A, 5L8B; EncFtn$_{sH}$-H65A, 5L8G (DOIs for X-ray diffraction image data are shown in *Table 4*). All MS datasets presented in this paper can be found, in the raw format at 10.7488/ds/1449.

## Acknowledgements

This work was supported by a Royal Society Research Grant awarded to JMW (RG130585), a BBSRC New Investigator Grant to JMW and DJC (BB/N005570/1) and a Wellcome Trust Institute Strategic Support Fund grant awarded to set up the XtalPod crystallization facility at the University of Edinburgh. DJC, SH, and KA are funded by the University of Edinburgh. DH is funded by the China Scholarship Council. KJW and ET are funded by the Wellcome Trust and Royal Society through a Sir Henry Dale Fellowship awarded to KJW (098375/Z/12/Z).

We would like to thank the staff on the Macromolecular Crystallography beamlines at Diamond Light Source for their assistance with data collection. We would like to thank Dr Steve Mitchell in the University of Edinburgh and Electron Microscopy Research Services in Newcastle University for the use of Transmission Electron Microscopes. We would like to thank Dr Michael Capeness and Dr. Louise Horsfall for the use of their anaerobic chamber. We would like to thank staff in the Edinburgh Protein Production Facility for their guidance and patience, Dr Janice Brahman, Dr Liz Blackburn, Dr Matthew Nowicki and Dr Martin Wear. We would like to thank Professor Rick Lewis and Dr Arnaud Basle (Newcastle University, UK), and Dr John Berrisford (PDBe, EMBL-EBI, Hinxton, Cambridge) for help identifying the glycolic acid ligand in the ferroxidase center. We would like to thank Prof Dominic Campopiano, Dr Elisabeth Lowe and Laura Tuck for their critical reading of this manuscript and helpful discussions.

## Additional information

### Funding

| Funder | Grant reference number | Author |
|---|---|---|
| China Scholarship Council | | Didi He |
| University Of Edinburgh | | Sam Hughes<br>Kirsten Altenbach<br>David J Clarke |
| Wellcome Trust | 098375/Z/12/Z | Emma Tarrant<br>Kevin J Waldron |
| Biotechnology and Biological Sciences Research Council | BB/N005570/1 | David J Clarke<br>Jon Marles-Wright |
| Royal Society | RG130585 | Jon Marles-Wright |

The funders had no role in study design, data collection and interpretation, or the decision to submit the work for publication.

### Author contributions

DH, DJC, JM-W, Conception and design, Acquisition of data, Analysis and interpretation of data, Drafting or revising the article; SH, SV-H, ET, Acquisition of data, Analysis and interpretation of data; AG, Acquisition of data, Drafting or revising the article; KA, CLM, Acquisition of data, Contributed unpublished essential data or reagents; KJW, Acquisition of data, Analysis and interpretation of data, Drafting or revising the article

Author ORCIDs

Didi He, http://orcid.org/0000-0002-3360-9352

Kevin J Waldron, http://orcid.org/0000-0002-5577-7357

David J Clarke, http://orcid.org/0000-0002-3741-2952

Jon Marles-Wright, http://orcid.org/0000-0002-9156-3284

## Additional files

### Supplementary files

• Supplementary file 1. Multiple sequence alignment of ferritin-like proteins. The amino acid sequences of ferritin family proteins were aligned progressively using EMBL-EBI web services (*Sutter et al., 2008*; *Akita et al., 2007*), Clustal Omega (*McHugh et al., 2014*) T-Coffee (*Contreras et al., 2014*) and MAFFT (*Helgstrand et al., 2003*). Protein names were adapted from either UniprotKB (*Sutter et al., 2008*), KEGG (*Roberts et al., 2011*) database or common name with PDB entry code (*He and Marles-Wright, 2015*). Sequences were sorted in an order corresponding to the clades in phylogenetic tree (*Figure 13*). The alignment was edited by Esprit 3.0 web server (*Aberg et al., 1993*). The *R. rubrum* EncFtn (Rru_A0973) sequence ishighlighted in yellow. The ferroxidase centre (FOC) of *Pseudo-nitzschia multiseries* ferritin (PmFtn_4ITW) (highlighting in blue) consists of $Fe_A$ site (E16, E49, E52) and $Fe_B$ site (E49, E95, E131, Q128) which are labelled with solid red triangles (*Grant et al., 1998*). Another iron binding site in PmFtn_4ITW (the gateway site or $Fe_C$ site [*Bradley et al., 2014*]) consists of E48, E45 and E131 which are marked with solid blue circles (*Grant et al., 1998*). The FOC of *R. rubrum* EncFtn is labelled with empty red triangles as E32, E62, H65 and Y39; and the iron entry site is marked with empty blue circles including E31 and E34. The putative iron exit site is marked with empty blue squares including H57, E61 and E64. The C-terminal localization sequences common to the encapsulin associated ferritins are highlighted within the red rectangle.

• Supplementary file 2. Primers used in this study. Primers used to generate the original constructs used in this study are listed 5' to 3', from left to right. Introduced restriction sites are shown underlined; regions complimentary to genomic DNA shown in bold.

### Major datasets

The following datasets were generated:

| Author(s) | Year | Dataset title | Dataset URL | Database, license, and accessibility information |
|---|---|---|---|---|
| Hughes S, Vanden-Hehir S, Clarke D | 2016 | Native MS Analysis of Encapsulated Ferritin from Rhodospirillum rubrum | http://dx.doi.org/10.7488/ds/1449 | Publicly available at Edinburgh DataShare |
| He D, Georgiev A, Marles-Wright J | 2016 | Single crystal X-ray diffraction data for Rhodospirillum rubrum encapsulated ferritin Rru_A0973, short His-tagged variant | http://dx.doi.org/10.7488/ds/1342 | Publicly available at Edinburgh DataShare |
| He D, Marles-Wright J | 2016 | Single crystal X-ray diffraction data for Rhodospirillum rubrum encapsulated ferritin Rru_A0973, short His-tagged variant E32A mutant | http://dx.doi.org/10.7488/ds/1419 | Publicly available at Edinburgh DataShare |
| He D, Marles-Wright J | 2016 | Single crystal X-ray diffraction data for Rhodospirillum rubrum encapsulated ferritin Rru_A0973, short His-tagged variant E62A mutant | http://dx.doi.org/10.7488/ds/1420 | Publicly available at Edinburgh DataShare |
| He D, Marles-Wright J | 2016 | Single crystal X-ray diffraction data for Rhodospirillum rubrum encapsulated ferritin Rru_A0973, short His-tagged variant H65A mutant | http://dx.doi.org/10.7488/ds/1421 | Publicly available at Edinburgh DataShare |

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
