## [Decision Letter]

[Editors’ note: a previous version of this study was rejected after peer review, but the authors submitted for reconsideration. The previous decision letter after peer review is shown below.]

Thank you for submitting your work entitled "Structural characterisation of an encapsulated ferritin provides insight into iron storage in bacterial nanocompartments" for consideration by *eLife*. Your article has been reviewed by two peer reviewers, and the evaluation has been overseen by a Reviewing Editor and Richard Losick as the Senior Editor. Our decision has been reached after consultation between the reviewers. Based on these discussions and the individual reviews below, we regret to inform you that your work will not be considered further for publication in *eLife*.

In this manuscript the authors characterize the iron binding and oxidation mechanisms of bacterial encapsulins. The widespread nature of these compartments and their potential physiological roles have only been appreciated recently, and thus represent an interesting frontier in microbial cell biology. While this study significantly advances our understanding of the structural and biochemical relationship between encapsulins and EncFer, it requires significant revision prior to publication. We do, however, encourage the authors to resubmit when and if they are address the issues raised below.

*Reviewer #1:*

1) Methods: What procedures and analyses did the author use to assess whether the iron added to the various ferritin derivatives was protein coated or was simply balls of rust attached to protein fragments? If the latter, it could easily generate reactive oxygen species in air under physiological conditions.

2) Results:

A) Critical data, such as the comparison of maximum amount of iron bound by a monomer in the dodocamer is in the Supplementary information.

B) The data in [Supplementary-material SD2-data], shows that the amount of iron bound by an ENCFTN decamer monomer is sub -stoichiometric, ranging from 0.18 to 0.64. In a bona fide ferritin, with ~ 2000 iron atoms/ protein cage (24 subunits), the same parameter is much, much higher.

Even an experimental situation: 24 subunit (monomer) ferritin with a biomineral prepared experimentally from apoferritin and containing, on average, only 1000 iron atoms/24 subunit cage, the equivalent parameter appears to be 1000/24 = 42. This Fe/protein ratio is 66 times more iron than in the test system described. Moreover, in nature, some ferritin protein cages contain as much as 4500 Fe atoms, several hundred times higher than the test system. Thus the significance of the experimental results in the paper are unclear.

3) Table 4: Data are shown for three proteins, Encapsulin, Enc-Ftn-10mer, and EncFTN-Enc. Missing are data for the starting material, 24 subunit ferritin or apoferritin (ferritin with the iron removed, by reduction and chelation, as a control.)

*Reviewer #2:*

In this manuscript the authors characterize the iron binding and oxidation mechanisms of bacterial encapsulins. The widespread nature of these compartments and their potential physiological roles have only been appreciated recently. While the structure of the encapsulin shell has been determined, that of its cargo, the ferritin-like protein (EncFer), has remained elusive. Here, the authors provide the structure of one such cargo and show that it assembles in a manner that is topologically distinct from ferritin. Additionally, the authors provide evidence that metal binding promotes the assembly of the EncFer and that it does act as a ferroxidase. Altogether, there is a substantial amount of work here that will likely be viewed as a major step forward in understanding these unique bacterial organelles. I have a few suggestions and questions that are listed below:

1) The authors grow *E. coli* in minimal media with and without added iron to show that assembly is iron dependent. The output of these experiments is the ratio of decamer vs. monomer. However, we don't have information on whether the growth conditions altered either the total amount of protein produced or the total amount soluble complex/monomer. Perhaps, lower protein concentrations lead to less efficient assembly (a critical concentration is needed).

2) There is no information regarding the reason for the use of *R. rubrum* encapsulins. As far as I can tell, these have not been a model for either in vivo or in vitro work. Is there even evidence that they are produced by *R. rubrum*? What is their size/appearance in that organism? Do they have a physiological role?

3) Also, how similar are the Enc and EncFer to those of *M. xanthus*? Are the putative iron-binding sites conserved?

4) I would have liked to see some mutagenesis experiments to test the models of assembly, iron binding and ferroxidase activity. These do not have to be in vivo and can be performed in vitro with the available system.

5) I would like some more phylogenetic data for the model that ferritin evolved from EncFer. Perhaps, EncFer evolved from ferritin? Do any of the existing phylogenetic analyses support one model over another.

---

## [Author Response]

[Editors’ note: the author responses to the first round of peer review follow.]

*In this manuscript the authors characterize the iron binding and oxidation mechanisms of bacterial encapsulins. The widespread nature of these compartments and their potential physiological roles have only been appreciated recently, and thus represent an interesting frontier in microbial cell biology. While this study significantly advances our understanding of the structural and biochemical relationship between encapsulins and EncFer, it requires significant revision prior to publication. We do, however, encourage the authors to resubmit when and if they are address the issues raised below.*

*Reviewer #1:*

*1) Methods: What procedures and analyses did the author use to assess whether the iron added to the various ferritin derivatives was protein coated or was simply balls of rust attached to protein fragments? If the latter, it could easily generate reactive oxygen species in air under physiological conditions.*

The reviewer makes an excellent point here. To ascertain whether the iron in the assays forms ‘balls of rust’ we performed transmission electron microscopy on the ferroxidase reaction mixtures after completion of the reaction to assess the formation of free, or encapsulated iron minerals. We provide an additional supplemental figure (Figure 8—figure supplement 1) and discuss the observation of iron mineral crystals and nanoparticles in the main text, subsection “Ferroxidase activity”, last paragraph. We also attempted to use a commercial luminescence-based ROS detection kit on the reactions to address the possibility that H_2_O_2_ is produced as a reaction intermediate by the EncFtn protein. We found that the results from this particular kit were inconsistent between repeats, but for the benefit of the reviewer we provide a graph of the results obtained (see Figure 15). These results show the production of ROS by apoferritin, which is consistent with the published data on the reaction mechanism of certain ferritins; however, no significant ROS were detected for the EncFtn or encapsulin proteins.

We acknowledge that the reaction mechanism of the EncFtn merits further investigation in a follow up study.10.7554/eLife.18972.037Author Response Image 1.**DOI:**
http://dx.doi.org/10.7554/eLife.18972.037

2) Results:

*A) Critical data, such as the comparison of maximum amount of iron bound by a monomer in the dodocamer is in the Supplementary information.*

We acknowledge that the data for iron loading merits inclusion in the main text, we have now moved this data and other supplementary data tables to the main text.

*B) The data in [Supplementary-material SD2-data], shows that the amount of iron bound by an ENCFTN decamer monomer is sub -stoichiometric, ranging from 0.18 to 0.64. In a bona fide ferritin, with ~ 2000 iron atoms/ protein cage (24 subunits), the same parameter is much, much higher.*

One of the central arguments of our paper is the fact that the EncFtnsH monomer must dimerize to produce a functional ferroxidase active site and that its iron binding properties are highly divergent from those of the classical ferritin nanocages. We have added additional text to the manuscript to highlight these differences (Introduction, last paragraph, and Mass spectrometry section) and discuss the functional consequences at length.

*Even an experimental situation: 24 subunit (monomer) ferritin with a biomineral prepared experimentally from apoferritin and containing, on average, only 1000 iron atoms/24 subunit cage, the equivalent parameter appears to be 1000/24 = 42. This Fe/protein ratio is 66 times more iron than in the test system described. Moreover, in nature, some ferritin protein cages contain as much as 4500 Fe atoms, several hundred times higher than the test system! Thus the significance of the experimental results in the paper are unclear.*

We have clarified this key difference in the discussion of the iron storage function of the encapsulin nanocompartment (subsection “Iron storage in encapsulin nanocompartments”, second paragraph). The key conclusion of the paper is that the iron storage and iron oxidation functions that are combined in classical ferritins are split between the encapsulin nanocompartment and the EncFtn protein.

*3) Table 4: Data are shown for three proteins, Encapsulin, Enc-Ftn-10mer, and EncFTN-Enc. Missing are data for the starting material, 24 subunit ferritin or apoferritin (ferritin with the iron removed, by reduction and chelation, as a control.)*

The data for the starting material are shown in Table 5. Control data for apoferritin have been added to this table and are illustrated in Figure 8. We note that we do not reach the experimental maximum loading capacity for apoferritin; however, we also note that the EncFtn-encapsulin nanocompartment sequesters five times more iron than the ferritin under the same reaction conditions, supporting the published observations that these nanocompartments can store more iron than classical ferritin nanocages.

*Reviewer #2:*

*In this manuscript the authors characterize the iron binding and oxidation mechanisms of bacterial encapsulins. The widespread nature of these compartments and their potential physiological roles have only been appreciated recently. While the structure of the encapsulin shell has been determined, that of its cargo, the ferritin-like protein (EncFer), has remained elusive. Here, the authors provide the structure of one such cargo and show that it assembles in a manner that is topologically distinct from ferritin. Additionally, the authors provide evidence that metal binding promotes the assembly of the EncFer and that it does act as a ferroxidase. Altogether, there is a substantial amount of work here that will likely be viewed as a major step forward in understanding these unique bacterial organelles. I have a few suggestions and questions that are listed below:*

*1) The authors grow E. coli in minimal media with and without added iron to show that assembly is iron dependent. The output of these experiments is the ratio of decamer vs. monomer. However, we don't have information on whether the growth conditions altered either the total amount of protein produced or the total amount soluble complex/monomer. Perhaps, lower protein concentrations lead to less efficient assembly (a critical concentration is needed).*

The reviewer makes an interesting point about growth conditions and we acknowledge that production of the protein in LB medium leads to varying protein yields and monomer/decamer proportions. We therefore adopted the use of M9 minimal medium throughout the study to give better reproducibility, which also enables better control of metal ion availability than the complex LB medium. Given the fact that the protein is produced recombinantly in *E. coli* it is not particularly instructive to prove the in vivoproduction of the EncFtn multimer in this host. We have added a panel to Figure 3 to show the effect of protein concentration on multimerization in vitro(Figure 3). Our mass spectrometry results show that the protein spontaneously multimerized in the presence of iron in vitroto form decameric species and that this is metal ion concentration dependent (Figure 7).

*2) There is no information regarding the reason for the use of R. rubrum encapsulins. As far as I can tell, these have not been a model for either in vivo or in vitro work. Is there even evidence that they are produced by R. rubrum? What is their size/appearance in that organism? Do they have a physiological role?*

We have put a comment in the Introduction to introduce *R. rubrum* (last paragraph). A preliminary study in the laboratory identified encapsulins in a preparation of lipid vesicles from *R. rubrum* containing chromatophores. We chose to follow up on these structures in this study. We do not feel this particular information is key to the central argument of the paper.

*3) Also, how similar are the Enc and EncFer to those of M. xanthus? Are the putative iron-binding sites conserved?*

We have noted this in the Introduction of the manuscript.

*4) I would have liked to see some mutagenesis experiments to test the models of assembly, iron binding and ferroxidase activity. These do not have to be* in vivo *and can be performed* in vitro *with the available system.*

To address this question we have produced three FOC mutants of the EncFtn protein and characterized these in solution, by mass spectrometry, and crystallographically (section: Mutagenesis of the EncFtnsH Ferroxidase center). We thank the reviewer for this suggestion as it highlighted the importance of the FOC residues for assembly and activity, and our new data has provided interesting insights into the EncFtn protein.

*5) I would like some more phylogenetic data for the model that ferritin evolved from EncFer. Perhaps, EncFer evolved from ferritin? Do any of the existing phylogenetic analyses support one model over another.*

We now include a phylogenetic tree (Figure 13) and consider the question of ferritin evolution in the Discussion (first paragraph). None of the authors of this study are evolutionary biologists but we appreciate the difficulty inherent in tracing the history of protein folds, especially in bacterial lineages. We refer to previous studies in this section and make a suggestion that can be followed up in subsequent studies.